# Sinkhorn Barycenters with Free Support via Frank-Wolfe Algorithm

**Giulia Luise**[1], **Saverio Salzo**[2], **Massimiliano Pontil**[1,2], **Carlo Ciliberto**[3]

`g.luise.16@ucl.ac.uk, saverio.salzo@iit.it, m.pontil@cs.ucl.ac.uk,c.ciliberto@ic.ac.uk`

[1] Department of Computer Science, University College London, UK
[2] CSML, Istituto Italiano di Tecnologia, Genova, Italy
[3] Department of Electrical and Electronic Engineering, Imperial College London, UK

## Abstract

We present a novel algorithm to estimate the barycenter of arbitrary probability distributions with respect to the Sinkhorn divergence. Based on a Frank-Wolfe optimization strategy, our approach proceeds by populating the support of the barycenter incrementally, without requiring any pre-allocation. We consider discrete as well as continuous distributions, proving convergence rates of the proposed algorithm in both settings. Key elements of our analysis are a new result showing that the Sinkhorn divergence on compact domains has Lipschitz continuous gradient with respect to the Total Variation and a characterization of the sample complexity of Sinkhorn potentials. Experiments validate the effectiveness of our method in practice.

## 1 Introduction

Aggregating and summarizing collections of probability measures is a key task in several machine learning scenarios. Depending on the metric adopted, the properties of the resulting average (or *barycenter*) of a family of probability measures vary significantly. By design, optimal transport metrics are better suited at capturing the geometry of the distribution than Euclidean distance or $f$-divergences [14]. In particular, Wasserstein barycenters have been successfully used in settings such as texture mixing [40], Bayesian inference [49], imaging [26], or model ensemble [18].

The notion of barycenter in Wasserstein space was first introduced by [2] and then investigated from the computational perspective for the original Wasserstein distance [12, 50, 54] as well as its entropic regularizations (e.g. Sinkhorn) [6, 14, 20]. Two main challenges in this regard are: $i$) how to efficiently identify the support of the candidate barycenter and $ii$) how to deal with continuous (or infinitely supported) probability measures. The first problem is typically addressed by either fixing the support of the barycenter a-priori [20, 50] or by adopting an alternating minimization procedure to iteratively optimize the support point locations and their weights [12, 14]. While fixed-support methods enjoy better theoretical guarantees, free-support algorithms are more memory efficient and practicable in high dimensional settings. The problem of dealing with continuous distributions has been mainly approached by adopting stochastic optimization methods to minimize the barycenter functional [12, 20, 50].

In this work we propose a novel method to compute the barycenter of a set of probability distributions with respect to the Sinkhorn divergence [25] that does not require to fix the support beforehand. We address both the cases of discrete and continuous probability measures. In contrast to previous free-support methods, our algorithm does not perform an alternate minimization between support and weights. Instead, we adopt a Frank-Wolfe (FW) procedure to populate the support by incrementally adding new points and updating their weights at each iteration, similarly to kernel herding strategies [5]. We prove the convergence of the proposed optimization scheme for both finitely and infinitely

supported distribution settings. A central result to our analysis is the characterization of regularity properties of Sinkhorn potentials (i.e., the dual solutions of the Sinkhorn divergence problem), which extends recent work in [21, 23]. We empirically evaluate the performance of the proposed algorithm.

**Contributions.** The analysis of the proposed algorithm hinges on the following contributions: $i$) we show that the gradient of the Sinkhorn divergence is Lipschitz continuous on the space of probability measures with respect to the Total Variation. This grants us convergence of the barycenter algorithm in finite settings. $ii$) We characterize the sample complexity of Sinkhorn potentials of two empirical distributions sampled from arbitrary probability measures. This latter result is interesting on its own but it also enables us to $iii$) design a concrete optimization scheme to approximately solve the barycenter problem for arbitrary probability measures with convergence guarantees. $iv$) A byproduct of our analysis is the generalization of the FW algorithm to settings where the objective functional is defined only on a set with empty interior, which is the case for Sinkhorn divergence barycenter problem.

The rest of the paper is organized as follows: Sec. 2 reviews standard notions of optimal transport theory. Sec. 3 introduces the barycenter functional, and analyses the Lipschitz continuity of its gradient. Sec. 4 describes the implementation of our algorithm and Sec. 5 studies its convergence rates. Finally, Sec. 6 evaluates the proposed methods empirically and Sec. 7 provides concluding remarks.

## 2 Background

The aim of this section is to recall definitions and properties of Optimal Transport theory with entropic regularization. Throughout the work, we consider a compact set $\mathcal{X} \subset \mathbb{R}^d$ and a symmetric cost function $\mathsf{c} \colon \mathcal{X} \times \mathcal{X} \to \mathbb{R}$. We set $\mathsf{D} := \sup_{x,y \in \mathcal{X}} \mathsf{c}(x,y)$ and denote by $\mathcal{M}_1^+(\mathcal{X})$ the space of probability measures on $\mathcal{X}$ (positive Radon measures with mass 1). For any $\alpha, \beta \in \mathcal{M}_1^+(\mathcal{X})$, the Optimal Transport problem with entropic regularization is defined as follow [13, 24, 38]

$$\mathrm{OT}_\varepsilon(\alpha, \beta) = \min_{\pi \in \Pi(\alpha,\beta)} \int_{\mathcal{X}^2} \mathsf{c}(x,y)\, d\pi(x,y) + \varepsilon \mathrm{KL}(\pi | \alpha \otimes \beta), \qquad \varepsilon \geq 0 \qquad (1)$$

where $\mathrm{KL}(\pi | \alpha \otimes \beta)$ is the *Kullback-Leibler divergence* between the candidate transport plan $\pi$ and the product distribution $\alpha \otimes \beta$, and $\Pi(\alpha, \beta) = \{\pi \in \mathcal{M}_+^1(\mathcal{X}^2) \colon \mathsf{P}_{1\#}\pi = \alpha,\ \mathsf{P}_{2\#}\pi = \beta\}$, with $\mathsf{P}_i \colon \mathcal{X} \times \mathcal{X} \to \mathcal{X}$ the projector onto the $i$-th component and $\#$ the push-forward operator. The case $\varepsilon = 0$ corresponds to the classic Optimal Transport problem introduced by Kantorovich [29]. In particular, if $\mathsf{c} = \|\cdot - \cdot\|^p$ for $p \in [1, \infty)$, then $\mathrm{OT}_0$ is the well-known $p$-Wasserstein distance [52]. Let $\varepsilon > 0$. Then, the dual problem of (1), in the sense of Fenchel-Rockafellar, is (see [10, 21])

$$\mathrm{OT}_\varepsilon(\alpha, \beta) = \max_{u,v \in \mathcal{C}(\mathcal{X})} \int u(x)\, d\alpha(x) + \int v(y)\, d\beta(y) - \varepsilon \int e^{\frac{u(x)+v(y)-\mathsf{c}(x,y)}{\varepsilon}}\, d\alpha(x) d\beta(y), \quad (2)$$

where $\mathcal{C}(\mathcal{X})$ denotes the space of real-valued continuous functions on $\mathcal{X}$, endowed with $\|\cdot\|_\infty$. Let $\mu \in \mathcal{M}_1^+(\mathcal{X})$. We denote by $\mathsf{T}_\mu \colon \mathcal{C}(\mathcal{X}) \to \mathcal{C}(\mathcal{X})$ the map such that, for any $w \in \mathcal{C}(\mathcal{X})$,

$$\mathsf{T}_\mu(w) \colon x \mapsto -\varepsilon \log \int e^{\frac{w(y)-\mathsf{c}(x,y)}{\varepsilon}}\, d\mu(y). \qquad (3)$$

The first order optimality conditions for (2) are (see [21] or Appendix B.2)

$$u = \mathsf{T}_\beta(v) \quad \alpha\text{- a.e.} \qquad \text{and} \qquad v = \mathsf{T}_\alpha(u) \quad \beta\text{- a.e.} \qquad (4)$$

Pairs $(u, v)$ satisfying (4) exist [30] and are referred to as *Sinkhorn potentials*. They are unique $(\alpha, \beta)$ - a.e. up to an additive constant, i.e., $(u + t, v - t)$ is also a solution for any $t \in \mathbb{R}$. In line with [21, 23] it will be useful in the following to assume $(u, v)$ to be the Sinkhorn potentials such that: $i$) $u(x_o) = 0$ for an arbitrary anchor point $x_o \in \mathcal{X}$ and $ii$) (4) is satisfied pointwise on the entire domain $\mathcal{X}$. Then, $u$ is a fixed point of the map $\mathsf{T}_{\beta\alpha} = \mathsf{T}_\beta \circ \mathsf{T}_\alpha$ (analogously for $v$). This suggests a fixed point iteration approach to minimize (2), yielding the well-known Sinkhorn-Knopp algorithm which has been shown to converge linearly in $\mathcal{C}(\mathcal{X})$ [30, 41]. See also Thm. B.10 for a precise statement. We recall a key result characterizing the differentiability of $\mathrm{OT}_\varepsilon$ in terms of the Sinkhorn potentials that will be useful in the following.

**Proposition 1** (Prop 2 in [21]). *Let* $\nabla\mathrm{OT}_\varepsilon\colon \mathcal{M}_1^+(\mathcal{X})^2 \to \mathcal{C}(\mathcal{X})^2$ *be such that,* $\forall \alpha, \beta \in \mathcal{M}_+^1(\mathcal{X})$

$$\nabla\mathrm{OT}_\varepsilon(\alpha,\beta) = (u,v), \qquad \text{with} \qquad u = \mathsf{T}_\beta(v),\ v = \mathsf{T}_\alpha(u)\ \text{on } \mathcal{X}, \quad u(x_o) = 0. \qquad (5)$$

*Then,* $\mathrm{OT}_\varepsilon$ *is directionally differentiable and,* $\forall \alpha, \alpha', \beta, \beta' \in \mathcal{M}_1^+(\mathcal{X})$, *the directional derivative of* $\mathrm{OT}_\varepsilon$ *at* $(\alpha,\beta)$ *along the feasible direction* $(\mu,\nu) = (\alpha' - \alpha, \beta' - \beta)$ *is*

$$\mathrm{OT}_\varepsilon'(\alpha,\beta;\mu,\nu) = \langle \nabla\mathrm{OT}_\varepsilon(\alpha,\beta), (\mu,\nu) \rangle = \langle u, \mu \rangle + \langle v, \nu \rangle, \qquad (6)$$

*where* $\langle w, \rho \rangle = \int w(x)\, d\rho(x)$ *denotes the canonical pairing between the spaces* $\mathcal{C}(\mathcal{X})$ *and* $\mathcal{M}(\mathcal{X})$.

Note that $\nabla\mathrm{OT}_\varepsilon$ is not a gradient in the standard sense. In particular note that the directional derivative in (6) is not defined for any pair of signed measures, but only along *feasible directions* $(\alpha'-\alpha, \beta'-\beta)$.

**Sinkhorn Divergence.** The fast convergence of Sinkhorn-Knopp algorithm makes $\mathrm{OT}_\varepsilon$ (with $\varepsilon > 0$) preferable to $\mathrm{OT}_0$ from a computational perspective [13]. However, when $\varepsilon > 0$ the entropic regularization introduces a bias in the optimal transport problem, since in general $\mathrm{OT}_\varepsilon(\mu,\mu) \neq 0$. To compensate for this bias, [25] introduced the Sinkhorn *divergence*

$$\mathsf{S}_\varepsilon\colon \mathcal{M}_1^+(\mathcal{X}) \times \mathcal{M}_1^+(\mathcal{X}) \to \mathbb{R}, \qquad (\alpha,\beta) \mapsto \mathrm{OT}_\varepsilon(\alpha,\beta) - \frac{1}{2}\mathrm{OT}_\varepsilon(\alpha,\alpha) - \frac{1}{2}\mathrm{OT}_\varepsilon(\beta,\beta), \qquad (7)$$

which was shown in [21] to be nonnegative, biconvex and to metrize the convergence in law under mild assumptions. We characterize the gradient of $\mathsf{S}_\varepsilon(\cdot,\beta)$ for a fixed $\beta \in \mathcal{M}_1^+(\mathcal{X})$, which will be key to derive our optimization algorithm for computing Sinkhorn barycenters.

**Remark 2.** *Let* $\nabla_1\mathrm{OT}_\varepsilon\colon \mathcal{M}_1^+(\mathcal{X})^2 \to \mathcal{C}(\mathcal{X})$ *denote the first component of* $\nabla\mathrm{OT}_\varepsilon$ *(informally the component* $u$ *of the Sinkhorn potentials* $(u,v)$*). Then, it follows from Prop. 1 and the definition of Sinkhorn divergence* (7) *that for any* $\beta \in \mathcal{M}_1^+(\mathcal{X})$ *the function* $\mathsf{S}_\varepsilon(\cdot,\beta)\colon \mathcal{M}_1^+(\mathcal{X}) \to \mathbb{R}$ *is directionally differentiable and admits gradient*

$$\nabla[\mathsf{S}_\varepsilon(\cdot,\beta)]\colon \mathcal{M}_1^+(\mathcal{X}) \to \mathcal{C}(\mathcal{X}) \qquad \alpha \mapsto \nabla_1\mathrm{OT}_\varepsilon(\alpha,\beta) - \frac{1}{2}\nabla_1\mathrm{OT}_\varepsilon(\alpha,\alpha) = u - p, \qquad (8)$$

*with* $u = \mathsf{T}_{\beta\alpha}(u)$ *and* $p = \mathsf{T}_{\alpha\alpha}(p)$ *the Sinkhorn potentials of* $\mathrm{OT}_\varepsilon(\alpha,\beta)$ *and* $\mathrm{OT}_\varepsilon(\alpha,\alpha)$ *respectively which are zero at* $x_o$.

We refer to Appendix C for an in-depth analysis of the directional differentiability properties of the Sinkorn divergence.

## 3 Sinkhorn barycenters with Frank-Wolfe

Given $\beta_1, \ldots \beta_m \in \mathcal{M}_1^+(\mathcal{X})$ and $\omega_1, \ldots, \omega_m \geq 0$ a set of weights such that $\sum_{j=1}^m \omega_j = 1$, the main goal of this paper is to solve the following *Sinkhorn barycenter* problem

$$\min_{\alpha \in \mathcal{M}_1^+(\mathcal{X})} \mathsf{B}_\varepsilon(\alpha), \qquad \text{with} \qquad \mathsf{B}_\varepsilon(\alpha) = \sum_{j=1}^m \omega_j\, \mathsf{S}_\varepsilon(\alpha,\beta_j). \qquad (9)$$

Although the objective functional $\mathsf{B}_\varepsilon$ is convex, its domain $\mathcal{M}_+^1(\mathcal{X})$ has *empty* interior in the space of finite signed measure $\mathcal{M}(\mathcal{X})$. Hence standard notions of Fréchet or Gâteaux differentiability do not apply. This, in principle causes some difficulties in devising optimization methods. To circumvent this issue, in this work we adopt the Frank-Wolfe (FW) algorithm. Indeed, one key advantage of this method is that it is formulated in terms of directional derivatives along feasible directions (i.e., directions that locally remain inside the constraint set). Building upon [15, 16, 19], which study the algorithm in Banach spaces, we show that the "weak" notion of directional differentiability of $\mathsf{S}_\varepsilon$ (and hence of $\mathsf{B}_\varepsilon$) in Remark 2 is sufficient to carry out the convergence analysis. While full details are provided in Appendix A, below we give an overview of the main result.

**Frank-Wolfe in dual Banach spaces.** Let $\mathcal{W}$ be a real Banach space with topological dual $\mathcal{W}^*$ and let $\mathcal{D} \subset \mathcal{W}^*$ be a nonempty, convex, closed and bounded set. For any $w \in \mathcal{W}^*$ denote by $\mathcal{F}_\mathcal{D}(w) = \mathbb{R}_+(\mathcal{D} - w)$ the set of feasible direction of $\mathcal{D}$ at $w$ (namely $s = t(w' - w)$ with $w' \in \mathcal{D}$ and $t > 0$). Let $\mathsf{G}\colon \mathcal{D} \to \mathbb{R}$ be a convex function and assume that there exists a map $\nabla\mathsf{G}\colon \mathcal{D} \to \mathcal{W}$ (not necessarily unique) such that $\langle \nabla\mathsf{G}(w), s \rangle = \mathsf{G}'(w; s)$ for every $s \in \mathcal{F}_\mathcal{D}(w)$. In Alg. 1 we present

---

**Algorithm 1** FRANK-WOLFE IN DUAL BANACH SPACES

---

**Input:** initial $w_0 \in \mathcal{D}$, precision $(\Delta_k)_{k \in \mathbb{N}} \in \mathbb{R}_{++}^{\mathbb{N}}$, such that $\Delta_k(k+2)$ is nondecreasing.

**For** $k = 0, 1, \ldots$

    Take $z_{k+1}$ such that $\mathsf{G}'(w_k, z_{k+1} - w_k) \leq \min_{z \in \mathcal{D}} \mathsf{G}'(w_k, z - w_k) + \frac{\Delta_k}{2}$

    $w_{k+1} = w_k + \frac{2}{k+2}(z_{k+1} - w_k)$

---

a method to minimize $\mathsf{G}$. The algorithm is structurally equivalent to the standard FW [19, 27] and accounts for possible inaccuracies when computing the conditional gradient (i.e. solving the FW inner minimization). This will be key in Sec. 5 when studying the barycenter problem for $\beta_j$ with infinite support. The following result (see proof in Appendix A) shows that under the additional assumption that $\nabla \mathsf{G}$ is Lipschitz-continuous and with sufficiently fast decay of the errors, the above procedure converges in value to the minimum of $\mathsf{G}$ with rate $O(1/k)$. Here $\operatorname{diam}(\mathcal{D})$ denotes the diameter of $\mathcal{D}$ with respect to the dual norm.

**Theorem 3.** *Under the assumptions above, suppose in addition that $\nabla \mathsf{G}$ is $L$-Lipschitz continuous with $L > 0$. Let $(w_k)_{k \in \mathbb{N}}$ and $(\Delta_k)_{k \in \mathbb{N}}$ be defined according to Alg. 1. Then, for every integer $k \geq 1$,*

$$\mathsf{G}(w_k) - \min_{w \in \mathcal{D}} \mathsf{G}(w) \leq \frac{2}{k+2} L \operatorname{diam}(\mathcal{D})^2 + \Delta_k. \tag{10}$$

**Frank-Wolfe Sinkhorn barycenters.** We show that the barycenter problem (9) satisfies the setting and hypotheses of Thm. 3 and can be thus approached via Alg. 1.

*Optimization domain.* Let $\mathcal{W} = \mathcal{C}(\mathcal{X})$, with dual $\mathcal{W}^* = \mathcal{M}(\mathcal{X})$. The constraint set $\mathcal{D} = \mathcal{M}_1^+(\mathcal{X})$ is convex, closed, and bounded.

*Objective functional.* The objective functional $\mathsf{G} = \mathsf{B}_\varepsilon : \mathcal{M}_1^+(\mathcal{X}) \to \mathbb{R}$, defined in (9), is convex since it is a convex combination of $\mathsf{S}_\varepsilon(\cdot, \beta_j)$, with $j = 1 \ldots m$. The gradient $\nabla \mathsf{B}_\varepsilon : \mathcal{M}_1^+(\mathcal{X}) \to \mathcal{C}(\mathcal{X})$ is $\nabla \mathsf{B}_\varepsilon = \sum_{j=1}^m \omega_j \nabla \mathsf{S}_\varepsilon(\cdot, \beta_j)$, where $\nabla \mathsf{S}_\varepsilon(\cdot, \beta_j)$ is given in Remark 2.

*Lipschitz continuity of the gradient.* This is the most critical condition and it is studied in the following theorem.

**Theorem 4.** *The gradient $\nabla \mathrm{OT}_\varepsilon$ defined in Prop. 1 is Lipschitz continuous. In particular, the first component $\nabla_1 \mathrm{OT}_\varepsilon$ is $2\varepsilon e^{3\mathsf{D}/\varepsilon}$-Lipschitz continuous, i.e., for every $\alpha, \alpha', \beta, \beta' \in \mathcal{M}_1^+(\mathcal{X})$,*

$$\|u - u'\|_\infty = \|\nabla_1 \mathrm{OT}_\varepsilon(\alpha, \beta) - \nabla_1 \mathrm{OT}_\varepsilon(\alpha', \beta')\|_\infty \leq 2\varepsilon e^{3\mathsf{D}/\varepsilon} \left( \|\alpha - \alpha'\|_{TV} + \|\beta - \beta'\|_{TV} \right), \tag{11}$$

*where $\mathsf{D} = \sup_{x,y \in \mathcal{X}} \mathsf{c}(x, y)$, $u = \mathsf{T}_{\beta\alpha}(u), u' = \mathsf{T}_{\beta'\alpha'}(u')$, and $u(x_o) = u'(x_o) = 0$. Moreover, it follows from (8) that $\nabla \mathsf{S}_\varepsilon(\cdot, \beta)$ is $6\varepsilon e^{3\mathsf{D}/\varepsilon}$-Lipschitz continuous. The same holds for $\nabla \mathsf{B}_\varepsilon$.*

Thm. 4 is one of the main contributions of this paper. It can be rephrased by saying that the operator that maps a pair of distributions to their Sinkhorn potentials is Lipschitz continuous. This result is significantly deeper than the one given in [20, Lemma 1], which establishes the Lipschitz continuity of the gradient in the *semidiscrete* case. The proof (given in Appendix D) relies on non-trivial tools from Perron-Frobenius theory for Hilbert's metric [32], which is a well-established framework to study Sinkhorn potentials [38]. We believe this result is interesting not only for the application of FW to the Sinkhorn barycenter problem, but also for further understanding regularity properties of entropic optimal transport.

## 4 Algorithm: practical Sinkhorn barycenters

According to Sec. 3, FW is a valid approach to tackle the barycenter problem (9). Here we describe how to implement in practice the abstract procedure of Alg. 1 to obtain a sequence of distributions $(\alpha_k)_{k \in \mathbb{N}}$ minimizing $\mathsf{B}_\varepsilon$. A main challenge in this sense resides in finding a minimizing feasible direction for $\mathsf{B}'_\varepsilon(\alpha_k; \mu - \alpha_k) = \langle \nabla \mathsf{B}_\varepsilon(\alpha_k), \mu - \alpha_k \rangle$. According to Remark 2, this amounts to solve

$$\mu_{k+1} \in \operatorname*{argmin}_{\mu \in \mathcal{M}_1^+(\mathcal{X})} \sum_{j=1}^m \omega_j \langle u_{jk} - p_k, \mu \rangle \qquad \text{where} \qquad u_{jk} - p_k = \nabla \mathsf{S}_\varepsilon[(\cdot, \beta_j)](\alpha_k), \tag{12}$$

---

**Algorithm 2** SINKHORN BARYCENTER

---

**Input:** $\beta_j = (\mathbf{Y}_j, \mathsf{b}_j)$ with $\mathbf{Y}_j \in \mathbb{R}^{d \times n_j}, \mathsf{b}_j \in \mathbb{R}^{n_j}, \omega_j > 0$ for $j = 1, \dots, m$, $x_0 \in \mathbb{R}^d, \varepsilon > 0, K \in \mathbb{N}$.

**Initialize:** $\alpha_0 = (\mathbf{X}_0, \mathsf{a}_0)$ with $\mathbf{X}_0 = x_0, \mathsf{a}_0 = 1$.

**For** $k = 0, 1, \dots, K - 1$

  $\mathsf{p} = \text{SINKHORNKNOPP}(\alpha_k, \alpha_k, \varepsilon)$
  $p(\cdot) = \text{SINKHORNGRADIENT}(\mathbf{X}_k, \mathsf{a}_k, \mathsf{p})$
  **For** $j = 1, \dots m$
      $\mathsf{v}_j = \text{SINKHORNKNOPP}(\alpha_k, \beta_j, \varepsilon)$
      $u_j(\cdot) = \text{SINKHORNGRADIENT}(\mathbf{Y}_j, \mathsf{b}_j, \mathsf{v}_j)$
  **Let** $\varphi \colon x \mapsto \sum_{j=1}^m \omega_j \, u_j(x) - p(x)$
  $x_{k+1} = \text{MINIMIZE}(\varphi)$
  $\mathbf{X}_{k+1} = [\mathbf{X}_k, x_{k+1}]$ and $\mathsf{a}_{k+1} = \frac{1}{k+2} [k \, \mathsf{a}_k, 2]$
  $\alpha_{k+1} = (\mathbf{X}_{k+1}, \mathsf{a}_{k+1})$

**Return:** $\alpha_K$

---

with $p_k = \nabla_1 \text{OT}_\varepsilon(\alpha_k, \alpha_k)$ not depending on $j$. In general (12) would entail a minimization over the set of all probability distributions on $\mathcal{X}$. However, since the objective functional is linear in $\mu$ and $\mathcal{M}_1^+(\mathcal{X})$ is a weakly-$*$ compact convex set, we can apply Bauer maximum principle (see e.g., [3, Thm. 7.69]). Hence, solutions are achieved at the extreme points of the optimization domain. These correspond to Dirac's deltas in the case of $\mathcal{M}_1^+(\mathcal{X})$ [11, p. 108]. Denote by $\delta_x \in \mathcal{M}_1^+(\mathcal{X})$ the Dirac's delta centered at $x \in \mathcal{X}$. We have $\langle w, \delta_x \rangle = w(x)$ for every $w \in \mathcal{C}(\mathcal{X})$. Hence (12) is equivalent to

$$\mu_{k+1} = \delta_{x_{k+1}} \qquad \text{with} \qquad x_{k+1} \in \underset{x \in \mathcal{X}}{\operatorname{argmin}} \sum_{j=1}^m \omega_j \left( u_{jk}(x) - p_k(x) \right). \tag{13}$$

Once the new support point $x_{k+1}$ has been obtained, the update in Alg. 1 corresponds to

$$\alpha_{k+1} = \alpha_k + \frac{2}{k+2}(\delta_{x_{k+1}} - \alpha_k) = \frac{k}{k+2}\alpha_k + \frac{2}{k+2}\delta_{x_{k+1}}. \tag{14}$$

If FW is initialized with a Dirac's delta $\alpha_0 = \delta_{x_0}$ for some $x_0 \in \mathcal{X}$, then every further iterate $\alpha_k$ will have at most $k + 1$ support points. According to (13), the inner optimization for FW consists in minimizing the functional $x \mapsto \sum_{j=1}^m \omega_j \left( u_{jk}(x) - p_k(x) \right)$ over $\mathcal{X}$. In practice, having access to such functional poses already a challenge, since it requires computing the Sinkhorn potentials $u_{jk}$ and $p_k$, which are infinite dimensional objects. Below we discuss how to estimate these potentials when the $\beta_j$ have finite support. We then address the general setting.

**Computing $\nabla_1 \text{OT}_\varepsilon$ for probability distributions with finite support.** Let $\alpha, \beta \in \mathcal{M}_1^+(\mathcal{X})$, where $\beta = \sum_{i=1}^n b_i \delta_{y_i}$ and $\mathsf{b} = (b_i)_{i=1}^n$ nonnegative weights summing up to 1. It is useful to identify $\beta$ with the pair $(\mathbf{Y}, \mathsf{b})$, where $\mathbf{Y} \in \mathbb{R}^{d \times n}$ is the matrix with $i$-th column equal to $y_i$. Let $(u, v) \in \mathcal{C}(\mathcal{X})^2$ be the pair of Sinkhorn potentials associated to $\alpha$ and $\beta$ in Prop. 1, recall that $u = \mathsf{T}_\beta(v)$. Denote by $\mathsf{v} \in \mathbb{R}^n$ the *evaluation vector* of the Sinkhorn potential $v$, with $i$-th entry $\mathsf{v}_i = v(y_i)$. According to the definition of $\mathsf{T}_\beta$ in (3), for any $x \in \mathcal{X}$

$$[\nabla_1 \text{OT}_\varepsilon(\alpha, \beta)](x) = u(x) = [\mathsf{T}_\beta(v)](x) = -\varepsilon \log \sum_{i=1}^n e^{(\mathsf{v}_i - \mathsf{c}(x, y_i))/\varepsilon} b_i, \tag{15}$$

since the integral $\mathsf{T}_\beta(v)$ reduces to a sum over the support of $\beta$. Hence, the gradient of $\text{OT}_\varepsilon$ (i.e. the potential $u$), *is uniquely characterized in terms of the finite dimensional vector $\mathsf{v}$ collecting the values of the potential $v$ on the support of $\beta$*. We refer as SINKHORNGRADIENT to the routine which associates to each triplet $(\mathbf{Y}, \mathsf{b}, \mathsf{v})$ the map $x \mapsto -\varepsilon \log \sum_{i=1}^n e^{(\mathsf{v}_i - \mathsf{c}(x, y_i))/\varepsilon} b_i$.

**Sinkhorn barycenters: finite case.** Alg. 2 summarizes FW applied to the barycenter problem (9) when the $\beta_j$'s have finite support. Starting from a Dirac's delta $\alpha_0 = \delta_{x_0}$, at each iteration $k \in \mathbb{N}$ the algorithm proceeds by: $i$) finding the corresponding evaluation vectors $\mathsf{v}_j$'s and $\mathsf{p}$ of the Sinkhorn potentials for $\text{OT}_\varepsilon(\alpha_k, \beta_j)$ and $\text{OT}_\varepsilon(\alpha_k, \alpha_k)$ respectively, via the routine SINKHORNKNOPP (see [13, 21] or Alg. B.2). This is possible since both $\beta_j$ and $\alpha_k$ have finite support and therefore the

problem of approximating the evaluation vectors $\mathsf{v}_j$ and $\mathsf{p}$ reduces to an optimization problem over finite vector spaces that can be efficiently solved [13]; $ii)$ obtain the gradients $u_j = \nabla_1 \mathrm{OT}_\varepsilon(\alpha_k, \beta_j)$ and $p = \nabla_1 \mathrm{OT}_\varepsilon(\alpha_j, \alpha_k)$ via SINKHORNGRADIENT; $iii)$ minimize $\varphi : x \mapsto \sum_{j=1}^n \omega_j \, u_j(x) - p(x)$ over $\mathcal{X}$ to find a new point $x_{k+1}$ (we comment on this meta-routine MINIMIZE below); $iv)$ finally update the support and weights of $\alpha_k$ according to (14) to obtain the new iterate $\alpha_{k+1}$.

A key feature of Alg. 2 is that the support of the candidate barycenter is updated *incrementally* by adding at most one point at each iteration, a procedure similar in flavor to the kernel herding strategy in [5, 31] and conditional gradient for sparse inverse problem [8, 9]. This contrasts with previous methods for barycenter estimation [6, 14, 20, 50], which require the support set, or at least its cardinality, to be fixed beforehand. However, indentifying the new support point requires solving the nonconvex problem (13), a task addressed by the meta-routine MINIMIZE. This problem is typically smooth (e.g., a linear combination of Gaussians when $\mathsf{c}(x, y) = \|x - y\|^2$) and first or second order nonlinear optimization methods can be adopted to find stationary points. We note that all free-support methods in the literature for barycenter estimation are also affected by nonconvexity since they typically require solving a biconvex problem (alternating minimization between support points and weights) which is not jointly convex [12, 14]. We conclude by observing that if we restrict to the setting of [20, 50] with fixed finite support set, then MINIMIZE can be solved exactly by evaluating the functional in (13) on each candidate support point.

**Sinkhorn barycenters: general case.** When the $\beta_j$'s have infinite support, it is not possible to apply Sinkhorn-Knopp in practice. In line with [23, 50], we can randomly sample empirical distributions $\hat{\beta}_j = \frac{1}{n} \sum_{i=1}^n \delta_{x_{ij}}$ from each $\beta_j$ and apply Sinkhorn-Knopp to $(\alpha_k, \hat{\beta}_j)$ in Alg. 1 rather than to the ideal pair $(\alpha_k, \beta_j)$. This strategy is motivated by [21, Prop 13], where it was shown that Sinkhorn potentials vary continuously with the input measures. However, it opens two questions: $i)$ whether this approach is theoretically justified (consistency) and $ii)$ how many points should we sample from each $\beta_j$ to ensure convergence (rates). We answer these questions in Thm. 7 in the next section.

## 5   Convergence analysis

We finally address the convergence of FW applied to both the finite and infinite settings discussed in Sec. 4. We begin by considering the finite setting.

**Theorem 5.** *Suppose that $\beta_1, \ldots \beta_m \in \mathcal{M}_1^+(\mathcal{X})$ have finite support and let $\alpha_k$ be the $k$-th iterate of Alg. 2 applied to (9). Then,*

$$\mathsf{B}_\varepsilon(\alpha_k) - \min_{\alpha \in \mathcal{M}_1^+(\mathcal{X})} \mathsf{B}_\varepsilon(\alpha) \leq \frac{48\,\varepsilon\,e^{3D/\varepsilon}}{k+2}. \tag{16}$$

The result follows by the convergence result of FW in Thm. 3 applied with the Lipschitz constant from Thm. 4, and recalling that $\mathrm{diam}(\mathcal{M}_1^+(\mathcal{X})) = 2$ with respect to the Total Variation. Note that Thm. 5 assumes SINKHORNKNOPP and MINIMIZE in Alg. 2 to yield exact solutions. In Appendix D we extend of Alg. 2 and Thm. 5 which account for approximation errors in the above routines.

**General setting.** As mentioned in Sec. 4, when the $\beta_j$'s are not finitely supported we adopt a sampling approach. More precisely we propose to *replace* in Alg. 2 the ideal Sinkhorn potentials of the pairs $(\alpha, \beta_j)$ with those of $(\alpha, \hat{\beta}_j)$, where each $\hat{\beta}_j$ is an empirical measure randomly sampled from $\beta_j$. In other words we are performing the FW algorithm with a (possibly rough) approximation of the correct gradient of $\mathsf{B}_\varepsilon$. According to Thm. 3, FW allows errors in the gradient estimation (which are captured into the precision $\Delta_k$ in the statement). To this end, the following result *quantifies* the approximation error between $\nabla_1 \mathrm{OT}_\varepsilon(\cdot, \beta)$ and $\nabla_1 \mathrm{OT}_\varepsilon(\cdot, \hat{\beta})$ in terms of the sample size of $\hat{\beta}$.

**Theorem 6** (Sample Complexity of Sinkhorn Potentials). *Suppose that $\mathsf{c} \in \mathcal{C}^{s+1}(\mathcal{X} \times \mathcal{X})$ with $s > d/2$. Then, there exists a constant $\bar{\mathsf{r}} = \bar{\mathsf{r}}(\mathcal{X}, \mathsf{c}, d)$ such that for any $\alpha, \beta \in \mathcal{M}_1^+(\mathcal{X})$ and any empirical measure $\hat{\beta}$ of a set of $n$ points independently sampled from $\beta$, we have, for every $\tau \in (0, 1]$*

$$\|u - u_n\|_\infty = \|\nabla_1 \mathrm{OT}_\varepsilon(\alpha, \beta) - \nabla_1 \mathrm{OT}_\varepsilon(\alpha, \hat{\beta})\|_\infty \leq \frac{8\varepsilon\,\bar{\mathsf{r}}e^{3D/\varepsilon} \log \frac{3}{\tau}}{\sqrt{n}} \tag{17}$$

*with probability at least $1 - \tau$, where $u = \mathsf{T}_{\beta\alpha}(u), u_n = \mathsf{T}_{\hat{\beta}\alpha}(u_n)$ and $u(x_o) = u_n(x_o) = 0$.*

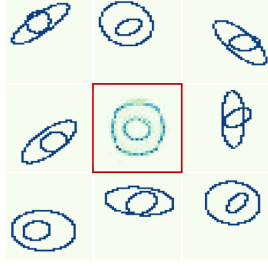
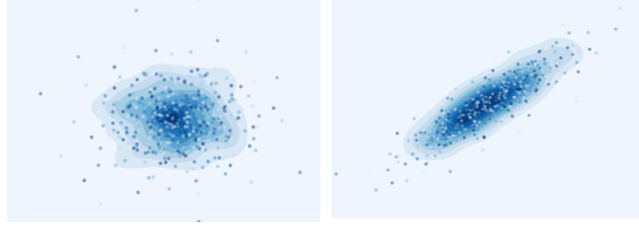

Fig. 1: Barycenter of nested ellipses      Fig. 2: Barycenters of Gaussians (see text)

The result in Thm. 6 is of central importance in this work. We point out that it *cannot* be obtained by means of the Lipschitz continuity of $\nabla_1 \mathrm{OT}_\varepsilon$ in Thm. 4, since empirical measures do not converge in $\|\cdot\|_{TV}$ to their target distribution [17]. Instead, the proof consists in considering the weaker *Maximum Mean Discrepancy (MMD)* metric associated to a universal kernel [46], which metrizes the topology of the convergence in law of $\mathcal{M}_1^+(\mathcal{X})$ [47]. Empirical measures converge in MMD metric to their target distribution [46]. Therefore, by proving the Lipschitz continuity of $\nabla_1 \mathrm{OT}_\varepsilon$ with *respect to* MMD (see Prop. E.5) we are able to conclude that (17) holds. This latter result relies on regularity properties of Sinkhorn potentials, which have been recently shown [23, Thm.2] to be uniformly bounded in Sobolev spaces under the additional assumption $c \in \mathcal{C}^{s+1}(\mathcal{X} \times \mathcal{X})$. For sufficiently large $s$, the Sobolev norm is in duality with the MMD [35] and allows us to derive the required Lipschitz continuity. We conclude noting that while [23] studied the sample complexity of the Sinkhorn *divergence*, Thm. 6 is a sample complexity result for Sinkhorn *potentials*. In this sense, we observe that the constants appearing in the bound are tightly related to those in [23, Thm.3] and have similar behavior with respect to $\varepsilon$. We can now study the convergence of FW in continuous settings.

**Theorem 7.** *Suppose that* $\mathsf{c} \in \mathcal{C}^{s+1}(\mathcal{X} \times \mathcal{X})$ *with* $s > d/2$. *Let* $n \in \mathbb{N}$ *and* $\hat{\beta}_1, \ldots, \hat{\beta}_m$ *be empirical distributions with* $n$ *support points, each independently sampled from* $\beta_1, \ldots, \beta_m$. *Let* $\alpha_k$ *be the* $k$-th *iterate of Alg. 2 applied to* $\hat{\beta}_1, \ldots, \hat{\beta}_m$. *Then for any* $\tau \in (0, 1]$, *the following holds with probability larger than* $1 - \tau$

$$\mathsf{B}_\varepsilon(\alpha_k) - \min_{\alpha \in \mathcal{M}_1^+(\mathcal{X})} \mathsf{B}_\varepsilon(\alpha) \leq \frac{64\bar{r}\varepsilon e^{3\mathrm{D}/\varepsilon} \log \frac{3m}{\tau}}{\min(k, \sqrt{n})}. \tag{18}$$

The proof is shown in Appendix E. A consequence of Thm. 7 is that the accuracy of FW depends simultaneously on the number of iterations and the sample size used in the approximation of the gradients: by choosing $n = k^2$ we recover the $O(1/k)$ rate of the finite setting, while for $n = k$ we have a rate of $O(k^{-1/2})$, which is reminiscent of typical sample complexity results, highlighting the statistical nature of the problem.

**Remark 8** (Incremental Sampling). *The above strategy requires sampling the empirical distributions for* $\beta_1, \ldots, \beta_m$ *beforehand. A natural question is whether it is be possible to do this* incrementally, *sampling points and updating* $\hat{\beta}_j$ *accordingly, as the number of* FW *iterations increase. To this end, one can perform an intersection bound and see that this strategy is still consistent, but the bound in Thm. 7 worsens the logarithmic term, which becomes* $\log(3mk/\tau)$.

## 6 Experiments

In this section we show the performance of our method in a range of experiments. Additional experiments are provided in the supplementary material. Code has been made publicly available[1].

**Discrete measures: barycenter of nested ellipses.** We compute the barycenter of 30 randomly generated nested ellipses on a $50 \times 50$ grid similarly to [14]. We interpret each image as a probability distribution in 2D. The cost matrix is given by the squared Euclidean distances between pixels. Fig. 1 reports 8 samples of the input ellipses and the barycenter obtained with Alg. 2. It shows qualitatively that our approach captures key geometric properties of the input measures.

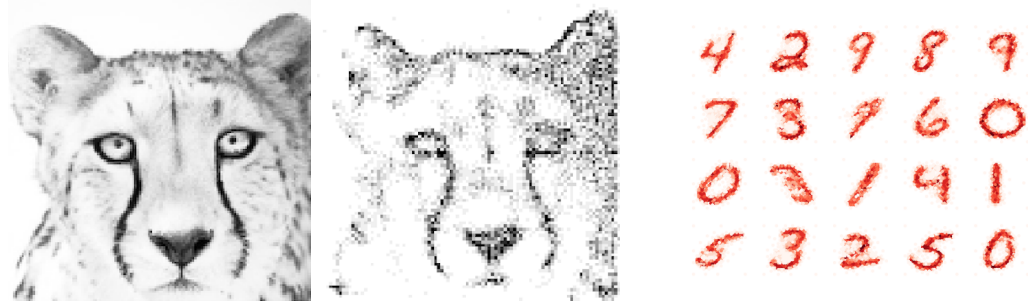

Fig. 3: Matching of a 140x140 image. 5000 FW iterations     Fig. 4: MNIST $k$-means (20 centers)

**Continuous measures: barycenter of Gaussians.** We compute the barycenter of 5 Gaussian distributions $\mathcal{N}(m_i, C_i)$ $i = 1, \ldots, 5$ in $\mathbb{R}^2$, with mean $m_i \in \mathbb{R}^2$ and covariance $C_i$ randomly generated. We apply Alg. 2 to empirical measures obtained by sampling $n = 500$ points from each $\mathcal{N}(m_i, C_i)$, $i = 1, \ldots, 5$. Since the (Wasserstein) barycenter of Gaussian distributions can be estimated accurately (see [2]), in Fig. 2 we report both the output of our method (as a scatter plot) and the true Wasserstein barycenter (as level sets of its density). We observe that our estimator recovers both the mean and covariance of the target barycenter. See the supplementary material for additional experiments also in the case of mixtures of Gaussians.

**Image "compression" via distribution matching.** Similarly to [12], we test Alg. 2 in the special case of computing the "barycenter" of a single measure $\beta \in \mathcal{M}_+^1(\mathcal{X})$. While the solution of this problem is the distribution $\beta$ itself, we can interpret the intermediate iterates $\alpha_k$ of Alg. 2 as compressed version of the original measure. In this sense $k$ would represent the level of compression since $\alpha_k$ is supported on *at most* $k$ points. Fig. 3 (Right) reports iteration $k = 5000$ of Alg. 2 applied to the $140 \times 140$ image in Fig. 3 (Left) interpreted as a probability measure $\beta$ in 2D. We note that the number of points in the support is $\sim 3900$: indeed, Alg. 2 selects the most relevant support points points multiple times to accumulate the right amount of mass on each of them (darker color = higher weight). This shows that FW tends to greedily search for the most relevant support points, prioritizing those with higher weight.

**k-means on MNIST digits.** We tested our algorithm on a $k$-means clustering experiment. We consider a subset of 500 random images from the MNIST dataset. Each image is suitably normalized to be interpreted as a probability distribution on the grid of $28 \times 28$ pixels with values scaled between 0 and 1. We initialize 20 centroids according to the $k$-means++ strategy [4]. Fig. 4 depicts the 20 centroids obtained by performing $k$-means with Alg. 2. We see that the structure of the digits is successfully detected, recovering also minor details (e.g. note the difference between the 2 centroids).

**Real data: Sinkhorn propagation of weather data.** We consider the problem of Sinkhorn *propagation* similar to the one in [45]. The goal is to predict the distribution of missing measurements for weather stations in the state of Texas, US by "propagating" measurements from neighboring stations in the network. The problem can be formulated as minimizing the functional $\sum_{(v,u)\in\mathcal{V}} \omega_{uv} \mathsf{S}_\varepsilon(\rho_v, \rho_u)$ over the set $\{\rho_v \in \mathcal{M}_1^+(\mathbb{R}^2) | v \in \mathcal{V}_0\}$ with: $\mathcal{V}_0 \subset \mathcal{V}$ the subset of stations with missing measurements, $G = (\mathcal{V}, \mathcal{E})$ the whole graph of the stations network, $\omega_{uv}$ a weight inversely proportional to the geographical distance between two vertices/stations $u, v \in \mathcal{V}$. The variable $\rho_v \in \mathcal{M}_1^+(\mathbb{R}^2)$ denotes the distribution of measurements at station $v$ of daily *temperature* and *atmospheric pressure* over one year. This is a generalization of the barycenter problem (9) (see also [38]).

From the total $|\mathcal{V}| = 115$, we randomly select $10\%, 20\%$ or $30\%$ to be *available* stations, and use Alg. 2 to propagate their measurements to the remaining "missing" ones. We compare our approach (FW) with the Dirichlet (DR) baseline in [45] in terms of the error $d(C_T, \hat{C})$ between the covariance matrix $C_T$ of the groundtruth distribution and that of the predicted one. Here $d(A, B) = \|\log(A^{-1/2} B A^{-1/2})\|$ is the geodesic distance on the cone of positive definite matrices. The average prediction errors are: 2.07 (FW), 2.24 (DR) for $10\%$, 1.47 (FW), 1.89(DR) for $20\%$ and 1.3 (FW), 1.6 (DR) for $30\%$. Fig. 5 qualitatively reports the improvement $\Delta = d(C_T, C_{DR}) - d(C_T, C_{FW})$ of our method on individual stations: a higher color intensity corresponds to a wider gap in our favor between prediction errors, from light green ($\Delta \sim 0$) to red ($\Delta \sim 2$). Our approach tends to propagate the distributions to missing locations with higher accuracy.

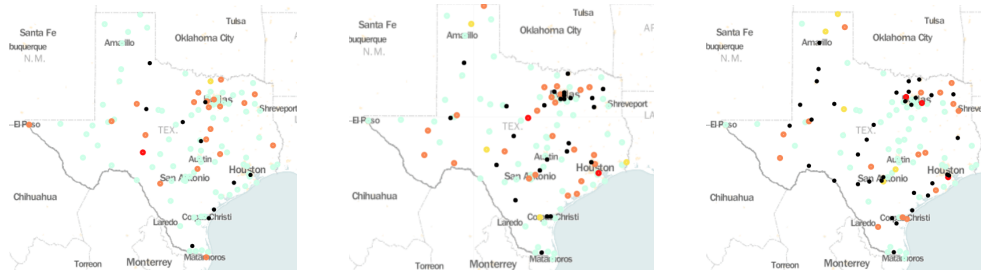

Fig. 5: From Left to Right: propagation of weather data with $10\%, 20\%$ and $30\%$ stations with available measurements (see text).

## 7    Conclusion

We proposed a Frank-Wolfe-based algorithm to find the Sinkhorn barycenter of probability distributions with either finitely or infinitely many support points. Our algorithm belongs to the family of barycenter methods with free support since it adaptively identifies support points rather than fixing them a-priori. In the finite settings, we were able to guarantee convergence of the proposed algorithm by proving the Lipschitz continuity of gradient of the barycenter functional in the Total Variation sense. Then, by studying the sample complexity of Sinkhorn potential estimation, we proved the convergence of our algorithm also in the infinite case. We empirically assessed our method on a number of synthetic and real experiments, showing that it exhibits good qualitative and quantitative performance. While in this work we have considered FW iterates that are a convex combination of Dirac's delta, models with higher regularity (e.g. mixture of Gaussians) might be more suited to approximate the barycenter of distributions with smooth density. Hence, in the future we plan to investigate whether the perspective adopted in this work could be extended also to other barycenter estimators.

## Footnotes

[1] https://github.com/GiulsLu/Sinkhorn-Barycenters

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
