[Supplementary Material]

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

# Supplementary Material

Below we give an overview of the structure of the supplementary material and highlight the main novel results of this work.

**Appendix A: abstract Frank-Wolfe algorithm in dual Banach spaces.** This section contains full details on Frank-Wolfe algorithm. The novelty stands in the relaxation of the differentiability assumptions.

**Appendix B: DAD problems and convergence of Sinkhorn-Knopp algorithm.** This section is a brief review of basic concepts from the nonlinear Perrom-Frobeius theory, DAD problems, and applications to the study of Sinkorn algorithm.

**Appendix C: Lipschitz continuitity of the gradient of the Sinkhorn divergence with respect to Total Variation.** This section contains one of the main contributions of our work, Theorem C.4, from which we derive Theorem 4 in the main text.

**Appendix D: Frank-Wolfe algorithm for Sinkhorn barycenters.** This section contains the complete analysis of FW algorithm for Sinkhorn barycenters, which takes into account the error in the computation of Sinkhorn potentials and the error in their minimization. The main result is the convergence of the Frank-Wolfe scheme for finitely supported distributions in Theorem D.2.

**Appendix E: Sample complexity of Sinkhorn potential and convergence of Algorithm 2 in case of continuous measures.** This section contains the discussion and the proofs of two of main results of the work Theorem 6, Theorem 7.

**Appendix F: additional experiments.** This section contains additional experiment on barycenters of mixture of Gaussian, barycenter of a mesh in 3D (dinosaur) and additional figures on the experiment on Sinkhorn propagation described in Section 6.

## A The Frank-Wolfe algorithm in dual Banach spaces

In this section we detail the convergence analysis of the Frank-Wolfe algorithm in abstract dual Banach spaces and under mild directional differentiablility assumptions so to cover the setting of Sinkhorn barycenters described in Section 3 of the paper.

Let $\mathcal{W}$ be a real Banach space and let $\mathcal{W}^*$ be its topological dual. Let $\mathcal{D} \subset \mathcal{W}^*$ be a nonempty, closed, convex, and bounded set and let $\mathsf{G} \colon \mathcal{D} \to \mathbb{R}$ be a convex function. We address the following optimization problem

$$\min_{w \in \mathcal{D}} \mathsf{G}(w), \tag{A.1}$$

assuming that the set of solutions is nonemtpy.

We recall the concept of the tangent cone of feasible directions.

**Definition A.1.** *Let* $w \in \mathcal{D}$. *Then* the cone of feasible directions *of* $\mathcal{D}$ *at* $w$ *is* $\mathcal{F}_{\mathcal{D}}(w) = \mathbb{R}_+(\mathcal{D} - w)$ *and the* tangent cone *of* $\mathcal{D}$ *at* $w$ *is*

$$\mathcal{T}_{\mathcal{D}}(w) = \overline{\mathcal{F}_{\mathcal{D}}(w)} = \big\{ v \in \mathcal{W}^* \,|\, (\exists (t_k)_{k \in \mathbb{N}} \in \mathbb{R}_{++}^{\mathbb{N}})(t_k \to 0)(\exists (w_k)_{k \in \mathbb{N}} \in \mathcal{D}^{\mathbb{N}}) \, t_k^{-1}(w_k - w) \to v \big\}.$$

**Remark A.1.** $\mathcal{F}_{\mathcal{D}}(w)$ is the cone generated by $\mathcal{D} - w$, and it is a convex cone. Indeed, if $t > 0$ and $v \in \mathcal{F}_{\mathcal{D}}(w)$, then $tv \in \mathcal{F}_{\mathcal{D}}(w)$. Moreover, if $v_1, v_2 \in \mathcal{F}_{\mathcal{D}}(w)$, then there exists $t_1, t_2 > 0$ and $w_1, w_2 \in \mathcal{D}$ such that $v_i = t_i(w_i - w)$, $i = 1, 2$. Thus,

$$v_1 + v_2 = (t_1 + t_2)\Big( \frac{t_1}{t_1 + t_2} w_1 + \frac{t_2}{t_1 + t_2} w_2 - w \Big) \in \mathbb{R}_+(\mathcal{D} - w).$$

So, $\mathcal{T}_{\mathcal{D}}(w)$ is a closed convex cone too.

**Definition A.2.** *Let $w \in \mathcal{D}$ and $v \in \mathcal{F}_\mathcal{D}(w)$. Then,* the directional derivative of $\mathsf{G}$ at $w$ in the direction $v$ *is*

$$\mathsf{G}'(w; v) = \lim_{t \to 0^+} \frac{\mathsf{G}(w + tv) - \mathsf{G}(w)}{t} \in [-\infty, +\infty[.$$

**Remark A.2.** The above definition is well-posed. Indeed, since $v$ is a feasible direction of $\mathcal{D}$ at $w$, there exists $t_1 > 0$ and $w_1 \in \mathcal{D}$ such that $v = t_1(w_1 - w)$; hence

$$(\forall\, t \in\, ]0, 1/t_1]) \quad w + tv = w + t\,t_1(w_1 - w) = (1 - t\,t_1)w + t\,t_1 w_1 \in \mathcal{D}.$$

Moreover, since $\mathsf{G}$ is convex, the function $t \in\, ]0, 1/t_1] \mapsto (\mathsf{G}(w+tv) - \mathsf{G}(w))/t$ is increasing, hence

$$\lim_{t \to 0^+} \frac{\mathsf{G}(w + tv) - \mathsf{G}(w)}{t} = \inf_{t \in ]0, 1/t_1]} \frac{\mathsf{G}(w + tv) - \mathsf{G}(w)}{t}. \tag{A.2}$$

It is easy to prove that the function

$$v \in \mathcal{F}_\mathcal{D}(w) \mapsto \mathsf{G}'(w; v) \in [-\infty, +\infty[$$

is positively homogeneous and sublinear (hence convex), that is,

   (i) $(\forall\, v \in \mathcal{F}_\mathcal{D}(w))(\forall\, t \in \mathbb{R}_+)\ \mathsf{G}'(w; tv) = t\mathsf{G}'(w; v)$;

   (ii) $(\forall\, v_1, v_2 \in \mathcal{F}_\mathcal{D}(w))\ \mathsf{G}'(w; v_1 + v_2) \le \mathsf{G}'(w; v_1) + \mathsf{G}'(w; v_2)$.

We make the following assumptions about $\mathsf{G}$:

   H1 $(\forall\, w \in \mathcal{D})$ the function $v \mapsto \mathsf{G}'(w; v)$ is finite, that is, $\mathsf{G}'(w; v) \in \mathbb{R}$.

   H2 The *curvature of* $\mathsf{G}$ is finite, that is,

$$C_\mathsf{G} = \sup_{\substack{w, z \in \mathcal{D} \\ \gamma \in [0,1]}} \frac{2}{\gamma^2}\big(\mathsf{G}(w + \gamma(z - w)) - \mathsf{G}(w) - \gamma\mathsf{G}'(w, z - w)\big) < +\infty. \tag{A.3}$$

**Remark A.3.** For every $w, z \in \mathcal{D}$, we have

$$\mathsf{G}(z) - \mathsf{G}(w) \ge \mathsf{G}'(w; z - w). \tag{A.4}$$

This follows from (A.2) with $w_1 = z$ and $t = 1$ ($t_1 = 1$).

The (inexact) Frank-Wolfe algorithm is detailed in Algorithm A.1.

---

**Algorithm A.1** Frank-Wolfe in Dual Banach Spaces

---

Let $(\gamma_k)_{k \in \mathbb{N}} \in \mathbb{R}_{++}^{\mathbb{N}}$ be such that $\gamma_0 = 1$ and, for every $k \in \mathbb{N}$, $1/\gamma_k \le 1/\gamma_{k+1} \le 1/2 + 1/\gamma_k$ (e.g., $\gamma_k = 2/(k+2)$). Let $w_0 \in \mathcal{D}$ and $(\Delta_k)_{k \in \mathbb{N}} \in \mathbb{R}_+^{\mathbb{N}}$ be such that $(\Delta_k/\gamma_k)_{k \in \mathbb{N}}$ is nondecreasing. Then

    for $k = 0, 1, \ldots$

       find $z_{k+1} \in \mathcal{D}$ is such that $\mathsf{G}'(w_k; z_{k+1} - w_k) \le \inf_{z \in \mathcal{D}} \mathsf{G}'(w_k; z - w_k) + \frac{1}{2}\Delta_k$

       $w_{k+1} = w_k + \gamma_k(z_{k+1} - w_k)$

---

**Remark A.4.**

   (i) Algorithm A.1 does not require the sub-problem $\min_{z \in \mathcal{D}} \mathsf{G}'(w_k; z - w_k)$ to have solutions. Indeed it only requires computing a $\Delta_k$-minimizer of $\mathsf{G}'(w_k; \cdot - w_k)$ on $\mathcal{D}$, which always exists.

   (ii) Since $\mathcal{D}$ is weakly-$*$ compact (by Banach-Alaoglu theorem), if $\mathsf{G}'(w_k; \cdot - w_k)$ is weakly-$*$ continuous on $\mathcal{D}$, then the sub-problem $\min_{z \in \mathcal{D}} \mathsf{G}'(w_k; z - w_k)$ admits solutions. Note that this occurs when the directional derivative $\mathsf{G}'(w; \cdot)$ is linear and can be represented in $\mathcal{W}$. This case is addressed in the subsequent Proposition A.7.

**Theorem A.5.** *Let $(w_k)_{k \in \mathbb{N}}$ be defined according to Algorithm A.1. Then, for every integer $k \ge 1$,*

$$\mathsf{G}(w_k) - \min_{w \in \mathcal{D}} \mathsf{G}(w) \le C_\mathsf{G}\gamma_k + \Delta_k. \tag{A.5}$$

*Proof.* Let $w_* \in \mathcal{D}$ be a solution of problem (A.1). It follows from H2 and the definition of $w_{k+1}$ in Algorithm A.1, that

$$\mathsf{G}(w_{k+1}) \leq \mathsf{G}(w_k) + \gamma_k \mathsf{G}'(w_k; z_{k+1} - w_k) + \frac{\gamma_k^2}{2} C_{\mathsf{G}}.$$

Moreover, it follows from the definition of $z_{k+1}$ in Algorithm A.1 and (A.4) that

$$\mathsf{G}'(w_k; z_{k+1} - w_k) \leq \inf_{z \in \mathcal{D}} \mathsf{G}'(w_k; z - w_k) + \frac{1}{2}\Delta_k$$

$$\leq \mathsf{G}'(w_k; w_* - w_k) + \frac{1}{2}\Delta_k$$

$$\leq -(\mathsf{G}(w_k) - \mathsf{G}(w_*)) + \frac{1}{2}\Delta_k.$$

Then,

$$\mathsf{G}(w_{k+1}) - \mathsf{G}(w_*) \leq (1 - \gamma_k)(\mathsf{G}(w_k) - \mathsf{G}(w_*)) + \frac{\gamma_k^2}{2}\left(C_{\mathsf{G}} + \frac{\Delta_k}{\gamma_k}\right). \tag{A.6}$$

Now, similarly to [28, Theorem 2], we can prove (A.5) by induction. Since $\gamma_0 = 1$, $1/\gamma_1 \leq 1/2 + 1/\gamma_0$, and $\Delta_0/\gamma_0 \leq \Delta_1/\gamma_1$, it follows from (A.6) that

$$\mathsf{G}(w_1) - \mathsf{G}(w_*) \leq \frac{1}{2}\left(C_{\mathsf{G}} + \frac{\Delta_0}{\gamma_0}\right) \leq \gamma_1\left(C_{\mathsf{G}} + \frac{\Delta_1}{\gamma_1}\right), \tag{A.7}$$

hence (A.5) is true for $k = 1$. Set, for the sake of brevity, $C_k = C_{\mathsf{G}} + \Delta_k/\gamma_k$ and suppose that (A.5) holds for $k \in \mathbb{N}$, $k \geq 1$. Then, it follows from (A.6) and the properties of $(\gamma_k)_{k \in \mathbb{N}}$ that

$$\mathsf{G}(w_{k+1}) - \mathsf{G}(w_*) \leq (1 - \gamma_k)\gamma_k C_k + \frac{\gamma_k^2}{2}C_k$$

$$= C_k \gamma_k\left(1 - \frac{\gamma_k}{2}\right)$$

$$\leq C_k \gamma_k\left(1 - \frac{\gamma_{k+1}}{2}\right)$$

$$\leq C_k \frac{1}{1/\gamma_{k+1} - 1/2}\left(1 - \frac{\gamma_{k+1}}{2}\right)$$

$$= C_k \gamma_{k+1}$$

$$\leq C_{k+1}\gamma_{k+1}. \qquad \square$$

**Corollary A.6.** *Under the assumptions of Theorem A.5, suppose in addition that $\Delta_k = \Delta\gamma_k^\zeta$, for some $\zeta \in [0, 1]$ and $\Delta \geq 0$. Then we have*

$$\mathsf{G}(w_k) - \min_{w \in \mathcal{D}} \mathsf{G}(w) \leq C_{\mathsf{G}}\gamma_k + \Delta\gamma_k^\zeta. \tag{A.8}$$

*Proof.* It follows from Theorem A.5 by noting that the sequence $\Delta_k/\gamma_k = 1/\gamma_k^{1-\zeta}$ is nondecreasing. $\square$

**Proposition A.7.** *Suppose that there exists a mapping $\nabla\mathsf{G}\colon \mathcal{D} \to \mathcal{W}$ such that[2],*

$$(\forall w \in \mathcal{D})(\forall z \in \mathcal{D}) \quad \langle \nabla\mathsf{G}(w), z - w \rangle = \mathsf{G}'(w; z - w). \tag{A.9}$$

*Then the following holds.*

(i) *Let $k \in \mathbb{N}$ and suppose that there exists $u_k \in \mathcal{W}$ such that $\|u_k - \nabla\mathsf{G}(w_k)\| \leq \Delta_{1,k}/4$ and that $z_{k+1} \in \mathcal{D}$ satisfies*

$$\langle u_k, z_{k+1} \rangle \leq \min_{z \in \mathcal{D}} \langle u_k, z \rangle + \frac{\Delta_{2,k}}{2},$$

*for some $\Delta_{1,k}, \Delta_{2,k} > 0$. Then*

$$\mathsf{G}'(w_k; z_{k+1} - w_k) \leq \min_{z \in \mathcal{D}} \mathsf{G}'(w_k; z - w_k) + \frac{1}{2}(\Delta_{1,k}\mathrm{diam}(\mathcal{D}) + \Delta_{2,k}). \tag{A.10}$$

(ii) *Suppose that $\nabla G \colon \mathcal{D} \to \mathcal{W}$ is L-Lipschitz continuous for some $L > 0$. Then, for every $w, z \in \mathcal{D}$ and $\gamma \in [0, 1]$,*

$$G(w + \gamma(z - w)) - G(w) - \gamma \langle z - w, \nabla G(w) \rangle \leq \frac{L}{2} \gamma^2 \|z - w\|^2$$

*and hence $C_G \leq L \mathrm{diam}(\mathcal{D})^2$.*

*Proof.* (i): We have

$$\langle \nabla G(w_k), z_{k+1} - w_k \rangle = \langle u_k, z_{k+1} - w_k \rangle + \langle \nabla G(w_k) - u_k, z_{k+1} - w_k \rangle$$

$$\leq \min_{z \in \mathcal{D}} \langle u_k, z - w_k \rangle + \frac{\Delta_{2,k}}{2} + \frac{\Delta_{1,k}}{4} \mathrm{diam}(\mathcal{D}). \qquad (A.11)$$

Moreover,

$$(\forall z \in \mathcal{D}) \quad \langle u_k, z - w_k \rangle = \langle \nabla G(w_k), z - w_k \rangle + \langle u_k - \nabla G(w_k), z - w_k \rangle$$

$$\leq \langle \nabla G(w_k), z - w_k \rangle + \frac{\Delta_{1,k}}{4} \mathrm{diam}(\mathcal{D}),$$

hence

$$\min_{z \in \mathcal{D}} \langle u_k, z - w_k \rangle \leq \min_{z \in \mathcal{D}} \langle \nabla G(w_k), z - w_k \rangle + \frac{\Delta_{1,k}}{4} \mathrm{diam}(\mathcal{D}). \qquad (A.12)$$

Thus, (A.10) follows from (A.11), (A.12), and (A.9).

(ii): Let $w, z \in \mathcal{D}$, and define $\psi \colon [0, 1] \to \mathcal{W}^*$ such that, $\forall \gamma \in [0, 1]$, $\psi(\gamma) = G(w + \gamma(z - w))$. Then, it is easy to see that for every $\gamma \in \,]0, 1[$, $\psi$ is differentiable at $\gamma$ and $\psi'(\gamma) = G'(w + \gamma(z - w); z - w) = \langle \nabla G(w + \gamma(z - w)), z - w \rangle$. Moreover, $\psi$ is continuous on $[0, 1]$. Therefore, the fundamental theorem of calculus yields

$$\psi(\gamma) - \psi(0) = \int_0^\gamma \psi'(t) dt$$

and hence

$$G(w + \gamma(z - w)) - G(w) - \langle \nabla G(w), z - w \rangle = \int_0^\gamma \langle \nabla G(w + t(z - w)) - \nabla G(w), z - w \rangle \, dt$$

$$\leq \int_0^\gamma \|\nabla G(w + t(z - w)) - \nabla G(w)\| \, \|z - w\| \, dt$$

$$\leq \int_0^\gamma Lt \|z - w\|^2 \, dt$$

$$= L \frac{\gamma^2}{2} \|z - w\|^2. \qquad \square$$

The following result is an extension of a classical result on the directional differentiability of a max function [7, Theorem 4.13] which relaxes the inf-compactness condition and allows the parameter space to be a convex set, instead of the entire Banach space. This result provides a prototype of functions (of which the entropic regularization of the Wasserstein distance is an instance) which are directionally differentiable only along the feasible directions of their domain and satisfies the hypotheses of Proposition A.7.

**Proposition A.8.** *Let $Z$ and $\mathcal{W}$ be real Banach spaces and let $\mathcal{W}^*$ be the topological dual of $\mathcal{W}$. Let $\mathcal{D} \subset \mathcal{W}^*$ be a nonempty closed convex set, and let $g \colon Z \times \mathcal{W}^* \to \mathbb{R}$ be such that*

1) *for every $z \in Z$, $g(z, \cdot) \colon \mathcal{W}^* \to \mathbb{R}$ is Gâteaux differentiable with derivative in $\mathcal{W}$, and the partial derivative with respect to the second variable $D_2 g \colon Z \times \mathcal{W}^* \to \mathcal{W}$ is continuous.*

2) *for every $w \in \mathcal{D}$, $S(w) := \mathrm{argmax}_Z \, g(\cdot, w) \neq \varnothing$.*

3) *there exists a continuous mapping $\varphi \colon \mathcal{D} \to Z$ such that, for every $w \in \mathcal{D}$, $\varphi(w) \in S(w)$.*

*Let* $\mathsf{G}\colon \mathcal{D} \to \mathbb{R}$ *be defined as*

$$\mathsf{G}(w) = \max_{z \in Z} g(z, w). \tag{A.13}$$

*Then,* $\mathsf{G}$ *is continuous, directionally differentiable, and, for every* $w \in \mathcal{D}$ *and* $v \in \mathcal{F}_{\mathcal{D}}(w)$

$$\mathsf{G}'(w; v) = \max_{z \in S(w)} \langle D_2 g(z, w), v \rangle = \langle D_2 g(\varphi(w), w), v \rangle. \tag{A.14}$$

*Proof.* The function $\mathsf{G}$ is well defined, since by assumption 2), for every $w \in \mathcal{D}$, $\operatorname{argmax}_Z g(\cdot, w) \neq \varnothing$. Let $w, u \in \mathcal{D}$ with $w \neq u$. Then, since $\varphi(w) \in S(w)$, we have $\mathsf{G}(w) = g(\varphi(w), w)$ and hence

$$\frac{\mathsf{G}(u) - \mathsf{G}(w) - \langle D_2 g(\varphi(w), w), u - w \rangle}{\|u - w\|}$$
$$\geq \frac{g(\varphi(w), u) - g(\varphi(w), w) - \langle D_2 g(\varphi(w), w), u - w \rangle}{\|u - w\|} \to 0, \quad \text{(A.15)}$$

since $g(\varphi(w), \cdot)$ is Fréchet differentiable[3] at $w$ with gradient $D_2 g(\varphi(w), w)$. Now, $\varphi(u) \in S(u)$, and hence $\mathsf{G}(u) = g(\varphi(u), u)$. Moreover, $g(\varphi(u), w) \leq \mathsf{G}(w)$. Therefore,

$$\frac{\mathsf{G}(u) - \mathsf{G}(w) - \langle D_2 g(\varphi(w), w), u - w \rangle}{\|u - w\|}$$
$$\leq \frac{g(\varphi(u), u) - g(\varphi(u), w) - \langle D_2 g(\varphi(w), w), u - w \rangle}{\|u - w\|}. \tag{A.16}$$

Let $\varepsilon > 0$. Since $D_2 g$ is continuous, there exists $\delta > 0$ such that, for every $z' \in Z$ and $w' \in \mathcal{W}^*$

$$\|z' - \varphi(w)\| \leq \delta \text{ and } \|w' - w\| \leq \delta \implies \|D_2 g(z', w') - D_2 g(\varphi(w), w)\| \leq \varepsilon. \tag{A.17}$$

Moreover, since $\varphi\colon \mathcal{D} \to Z$ is continuous, there exists $\eta > 0$ such that,

$$\|u - w\| \leq \eta \implies \|\varphi(u) - \varphi(w)\| \leq \delta. \tag{A.18}$$

Let $z' \in Z$ and suppose that $\|z' - \varphi(w)\| \leq \delta$ and $\|u - w\| \leq \delta$. Define $\psi\colon [0, 1] \to \mathbb{R}$ such that, for every $s \in [0, 1]$, $\psi(s) = g(z', w + s(u - w))$. Then, $\psi$ is continuously differentiable on $[0, 1]$ and $\psi'(s) = \langle D_2 g(z', w + s(u - w)), u - w \rangle$. Therefore,

$$\psi(1) - \psi(0) = \int_0^1 \psi'(s) ds \tag{A.19}$$

and hence, it follows from (A.17) that

$$|g(z', u) - g(z', w) - \langle D_2 g(\varphi(w), w), u - w \rangle|$$
$$= \left| \int_0^1 \langle D_2 g(z', w + s(u - w)) - D_2 g(\varphi(w), w), u - w \rangle \, ds \right|$$
$$\leq \int_0^1 \|D_2 g(z', w + s(u - w)) - D_2 g(\varphi(w), w)\| \, \|u - w\| \, ds$$
$$\leq \varepsilon \|u - w\|.$$

Therefore, we derive from (A.18), that for every $u \in \mathcal{D}$ such that $\|u - w\| \leq \min\{\eta, \delta\}$, we have

$$\left| \frac{g(\varphi(u), u) - g(\varphi(u), w) - \langle D_2 g(\varphi(w), w), u - w \rangle}{\|u - w\|} \right| \leq \varepsilon.$$

This shows that

$$\lim_{\substack{u \in \mathcal{D} \\ u \to w}} \frac{g(\varphi(u), u) - g(\varphi(u), w) - \langle D_2 g(\varphi(w), w), u - w \rangle}{\|u - w\|} = 0. \tag{A.20}$$

Then, we derive from (A.15), (A.16), and (A.20) that

$$\lim_{\substack{u \in \mathcal{D} \\ u \to w}} \frac{\mathsf{G}(u) - \mathsf{G}(w) - \langle D_2 g(\varphi(w), w), u - w \rangle}{\|u - w\|} = 0. \tag{A.21}$$

This implies that $\lim_{u\in\mathcal{D}, u\to w} \mathsf{G}(u) = \mathsf{G}(w)$. Moreover, if $v \in \mathcal{F}_{\mathcal{D}}(w)$, there exists $\lambda > 0$ and $u \in \mathcal{D}$ such that $v = \lambda(u - w)$ and, for every $t \in ]0, 1/\lambda]$,

$$
\frac{\mathsf{G}(w + tv) - \mathsf{G}(w)}{t} - \langle D_2 g(\varphi(w), w), v \rangle
$$
$$
= \|\lambda(u - w)\| \frac{\mathsf{G}(w + t\lambda(u - w)) - \mathsf{G}(w) - \langle D_2 g(\varphi(w), w), t\lambda(u - w)\rangle}{\|t\lambda(u - w)\|} \quad \text{(A.22)}
$$

and the right hand side goes to zero as $t \to 0^+$, because of (A.21). Therefore, for every $z \in S(w)$, since $\mathsf{G}(w) = g(z, w)$ and $\mathsf{G}(w + tv) \geq g(z, w + tv)$, we have

$$
\langle D_2 g(\varphi(w), w), v \rangle = \lim_{t\to 0^+} \frac{\mathsf{G}(w + tv) - \mathsf{G}(w)}{t} \geq \lim_{t\to 0^+} \frac{g(z, w + tv) - g(z, w)}{t} = \langle D_2 g(z, w), v \rangle
$$

and (A.14) follows. $\qquad\square$

# B   DAD problems and convergence of Sinkhorn-Knopp algorithm

In this section we review the basic concepts of the nonlinear Perron-Frobenius theory [32] which provides tools for dealing with DAD problems and ultimately to study the key properties of the Sinkhorn potentials. This analysis will allow us to provide in Appendix C an upper bound estimate for the Lipschitz constant of the gradient of $\mathsf{B}_\varepsilon$, which is needed in the Frank-Wolfe algorithm.

## B.1   Hilbert's metric and the Birkhoff-Hopf theorem

In the rest of the appendix we will assume $\mathcal{X} \subset \mathbb{R}^d$ to be a compact set. We denote by $\mathcal{C}(\mathcal{X})$ the space of continuous functions on $\mathcal{X}$ endowed with the sup norm, namely $\|f\|_\infty = \sup_{x\in\mathcal{X}} |f(x)|$. Let $\mathcal{C}_+(\mathcal{X})$ be the cone of nonnegative continuous functions, that is, $f \in \mathcal{C}(\mathcal{X})$ such that $f(x) \geq 0$ for every $x \in \mathcal{X}$. Also, we denote by $\mathcal{C}_{++}(\mathcal{X})$ the set of continuous and (strictly) positive functions on $\mathcal{X}$, which turns out to be the interior of $\mathcal{C}_+(\mathcal{X})$.

Let $\mathsf{c} : \mathcal{X} \times \mathcal{X} \to \mathbb{R}_+$ be a positive, symmetric, and continuous function and define $\mathsf{k} : \mathcal{X} \times \mathcal{X} \to \mathbb{R}_{++}$ as

$$
(\forall x, y \in \mathcal{X}) \qquad \mathsf{k}(x, y) = e^{-\frac{\mathsf{c}(x,y)}{\varepsilon}}. \tag{B.1}
$$

Set $\mathsf{D} = \sup_{x,y\in\mathcal{X}} \mathsf{c}(x, y)$. Then, we have $\mathsf{k}(x, y) \in [e^{-\mathsf{D}/\varepsilon}, 1]$ for all $x, y \in \mathcal{X}$. Let $\alpha \in \mathcal{M}_1^+(\mathcal{X})$. The operator $\mathsf{L}_\alpha \colon \mathcal{C}(\mathcal{X}) \to \mathcal{C}(\mathcal{X})$ is defined as

$$
(\forall f \in \mathcal{C}(\mathcal{X})) \qquad \mathsf{L}_\alpha f \colon x \mapsto \int \mathsf{k}(x, z) f(z)\, d\alpha(z). \tag{B.2}
$$

Note that $\mathsf{L}_\alpha$ is linear and continuous. In particular, since $k(x, y) \in [0, 1]$ for all $x, y \in \mathcal{X}$, we have

$$
(\forall\, f \in \mathcal{C}_+(\mathcal{X})) \qquad \mathsf{L}_\alpha f \geq 0 \tag{B.3}
$$

and

$$
(\forall\, f \in \mathcal{C}(\mathcal{X})) \qquad \|\mathsf{L}_\alpha f\|_\infty \leq \|f\|_\infty. \tag{B.4}
$$

**Hilbert's Metric.** The cone $\mathcal{C}_+(\mathcal{X})$ induces a partial ordering $\leq$ on $\mathcal{C}(\mathcal{X})$, such that

$$
(\forall\, f, f' \in \mathcal{C}(\mathcal{X})) \qquad f \leq f' \Leftrightarrow f' - f \in \mathcal{C}_+(\mathcal{X}). \tag{B.5}
$$

According to [32], we say that a function $f' \in \mathcal{C}_+(\mathcal{X})$ *dominates* $f \in \mathcal{C}(\mathcal{X})$ if there exist $t, s \in \mathbb{R}$ such that

$$
tf' \leq f \leq sf'. \tag{B.6}
$$

This notion induces an equivalence relation on $\mathcal{C}_+(\mathcal{X})$, denoted $f \sim f'$, meaning that $f$ dominates $f'$ and $f'$ dominates $f$. The corresponding equivalence classes are called *parts* of $\mathcal{C}_+(\mathcal{X})$. Let $f, f' \in \mathcal{C}_+(\mathcal{X})$ be such that $f \sim f'$. We define

$$
M(f/f') = \inf\{s \in \mathbb{R} \,|\, f \leq sf'\} \qquad \text{and} \qquad m(f/f') = \sup\{t \in \mathbb{R} \,|\, tf' \leq f\}. \tag{B.7}
$$

Note that $m(f/f') \leq M(f/f')$. Moreover, for every $f, f' \in \mathcal{C}_+(\mathcal{X})$ such that $f \sim f'$, we have that $\text{supp}(f) = \text{supp}(f')$ and if $f' \neq 0$ (hence $f \neq 0$), then

$$M(f/f') = \max_{x \in \text{supp}(f')} \frac{f(x)}{f'(x)} > 0 \quad \text{and} \quad m(f/f') = \min_{x \in \text{supp}(f')} \frac{f(x)}{f'(x)} > 0. \qquad \text{(B.8)}$$

The *Hilbert's metric* is defined as

$$d_H(f, f') = \log \frac{M(f/f')}{m(f/f')}, \qquad \text{(B.9)}$$

for all $f \sim f'$ with $f \neq 0$ and $f' \neq 0$, $d_H(0,0) = 0$ and $d_H(f, f') = +\infty$ otherwise. Direct calculation shows that [33, Proposition 2.1.1]

(i) $d_H(f, f') \geq 0$ and $d_H(f, f') = d_H(f', f)$, for every $f, f' \in \mathcal{C}_+(\mathcal{X})$;

(ii) $d_H(f, f'') \leq d_H(f, f') + d_H(f', f'')$, for every $f, f', f'' \in \mathcal{C}_+(\mathcal{X})$ with $f \sim f'$ and $f' \sim f''$;

(iii) $d_H(sf, tf') = d_H(f, f')$, for every $f, f' \in \mathcal{C}_+(\mathcal{X})$ and $s, t > 0$.

Note that $d_H$ is not a metric on the parts of $\mathcal{C}_+(\mathcal{X})$. However the set $\mathcal{C}_{++}(\mathcal{X}) \cap \partial B_1(0) = \{f \in \mathcal{C}_{++}(\mathcal{X}) \mid \|f\|_\infty = 1\}$ equipped with $d_H$ is a complete metric space [36]. Also, $d_H$ induces a metric on the rays of the parts of $\mathcal{C}_+(\mathcal{X})$ [33, Lemma 2.1].

We now focus on $\mathcal{C}_{++}(\mathcal{X})$. A direct consequence of Hilbert's metric properties is the following.

**Lemma B.1** (Hilbert's Metric on $\mathcal{C}_{++}(\mathcal{X})$)**.** *The interior of $\mathcal{C}_+(\mathcal{X})$ corresponds to the set of (strictly) positive functions $\mathcal{C}_{++}(\mathcal{X})$ and is a part of $\mathcal{C}_+(\mathcal{X})$ with respect to the equivalence relation induced by dominance. For every $f, f' \in \mathcal{C}_{++}(\mathcal{X})$,*

$$M(f/f') = \max_{x \in \mathcal{X}} \frac{f(x)}{f'(x)} \qquad m(f/f') = \min_{x \in \mathcal{X}} \frac{f(x)}{f'(x)}, \qquad \text{(B.10)}$$

*and $M(f/f') \geq m(f/f') > 0$. Therefore*

$$d_H(f, f') = \log \max_{x,y \in \mathcal{X}} \frac{f(x) \, f'(y)}{f(y) \, f'(x)}. \qquad \text{(B.11)}$$

*Proof.* Since $\mathcal{X}$ is compact it is straightfoward to see that $\mathcal{C}_{++}(\mathcal{X})$ is the interior of $\mathcal{C}_+(\mathcal{X})$. By applying [32, Lemma 1.2.2] we have that $\mathcal{C}_{++}(\mathcal{X})$ is a part of $\mathcal{C}_+(\mathcal{X})$. The characterization of $M(f/f')$ and $m(f/f')$ follow by direct calculation from the definition using the fact that $\inf_\mathcal{X} h = \min_\mathcal{X} h > 0$ for any $h \in \mathcal{C}_{++}(\mathcal{X})$ since $\mathcal{X}$ is compact. Finally, the characterization of Hilbert's metric on $\mathcal{C}_{++}(\mathcal{X})$ is obtained by recalling that $(\min_{x \in \mathcal{X}} h(x))^{-1} = \max_{x \in \mathcal{X}} h(x)^{-1}$ for every $h \in \mathcal{C}_{++}(\mathcal{X})$. $\qquad \square$

**Lemma B.2** (Ordering properties of $\mathsf{L}_\alpha$)**.** *Let $\alpha \in \mathcal{M}_1^+(\mathcal{X})$. Then the following holds:*

(i) *the operator $\mathsf{L}_\alpha$ is order-preserving (with respect to the cone $\mathcal{C}_+(\mathcal{X})$), that is,*

$$(\forall f, f' \in \mathcal{C}(\mathcal{X})) \qquad f \leq f' \Rightarrow \mathsf{L}_\alpha f \leq \mathsf{L}_\alpha f'; \qquad \text{(B.12)}$$

(ii) *$\mathsf{L}_\alpha$ maps parts of $\mathcal{C}_+(\mathcal{X})$ into parts of $\mathcal{C}_+(\mathcal{X})$, that is,*

$$(\forall f, f' \in \mathcal{C}_+(\mathcal{X})) \qquad f \sim f' \Rightarrow \mathsf{L}_\alpha f \sim \mathsf{L}_\alpha f'; \qquad \text{(B.13)}$$

(iii) *$\mathsf{L}_\alpha(\mathcal{C}_+(\mathcal{X})) \subset \mathcal{C}_{++}(\mathcal{X}) \cup \{0\}$ and $\mathsf{L}_\alpha(\mathcal{C}_{++}(\mathcal{X})) \subset \mathcal{C}_{++}(\mathcal{X})$.*

*Proof.* (i): Let $f, f' \in \mathcal{C}(\mathcal{X})$ with $f \leq f'$. Then $f' - f \in \mathcal{C}_+(\mathcal{X})$ and by linearity of $\mathsf{L}_\alpha$ combined with (B.3), we have $\mathsf{L}_\alpha f' - \mathsf{L}_\alpha f = \mathsf{L}_\alpha(f - f') \geq 0$.

(ii): Let $f, f' \in \mathcal{C}_+(\mathcal{X})$ with $f \sim f'$. Then there exist $t, s \in \mathbb{R}$ and $s', t' \in \mathbb{R}$ such that $tf' \leq f \leq sf'$ and $t'f \leq f' \leq s'f$. Since $L_\alpha$ is linear and order-preserving, we have $\mathsf{L}_\alpha f \sim \mathsf{L}_\alpha f'$.

(iii): Let $f \in \mathcal{C}_+(\mathcal{X})$. By (B.3) and (B.4), for any $x \in \mathcal{X}$

$$0 \leq (\mathsf{L}_\alpha f)(x) \leq \|\mathsf{L}_\alpha f\|_\infty \leq \int f(x) \, d\alpha(x) = \|f\|_{L^1(\mathcal{X}, \alpha)}. \qquad \text{(B.14)}$$

Moreover,

$$\mathsf{L}_\alpha f(x) = \int k(y,x) f(y)\, d\alpha(y) \geq e^{-\mathsf{D}/\varepsilon}\, \|f\|_{L^1(\mathcal{X},\alpha)}. \tag{B.15}$$

Therefore, if $\|f\|_{L^1(\mathcal{X},\alpha)} = 0$ then by (B.14) $\mathsf{L}_\alpha f = 0$ while, if $\|f\|_{L^1(\mathcal{X},\alpha)} > 0$ then by (B.15) $\mathsf{L}_\alpha f \in \mathcal{C}_{++}(\mathcal{X})$. We conclude that the operator $\mathsf{L}_\alpha$ maps $\mathcal{C}_+(\mathcal{X})$ in $\mathcal{C}_{++}(\mathcal{X}) \cup \{0\}$. Moreover, $\mathsf{L}_\alpha(\mathcal{C}_{++}(\mathcal{X})) \subset \mathcal{C}_{++}(\mathcal{X})$, since for every $f \in \mathcal{C}_{++}(\mathcal{X})$ we have $\|f\|_{L^1(\mathcal{X},\alpha)} \geq \min_\mathcal{X} f > 0$. $\qquad\square$

Following [32, Section A.4] we now introduce a quantity which plays a central role in our analysis.

**Definition B.1** (Projective Diameter of $\mathsf{L}_\alpha$). *Let $\alpha \in \mathcal{M}_1^+(\mathcal{X})$. The* projective diameter *of $\mathsf{L}_\alpha$ is*

$$\Delta(\mathsf{L}_\alpha) = \sup\{d_H(\mathsf{L}_\alpha f, \mathsf{L}_\alpha f') \mid f, f' \in \mathcal{C}_+(\mathcal{X}),\ \mathsf{L}_\alpha f \sim \mathsf{L}_\alpha f'\}. \tag{B.16}$$

The following result shows that it is possible to find a finite upper bound on $\Delta(\mathsf{L}_\alpha)$ that is independent on $\alpha$.

**Proposition B.3** (Upper bound on the Projective Diameter of $\mathsf{L}_\alpha$). *Let $\alpha \in \mathcal{M}_1^+(\mathcal{X})$. Then*

$$\Delta(\mathsf{L}_\alpha) \leq 2\mathsf{D}/\varepsilon. \tag{B.17}$$

*Proof.* Let $f, f' \in \mathcal{C}_+(\mathcal{X})$. Recall that $\mathsf{L}_\alpha$ maps $\mathcal{C}_+(\mathcal{X})$ into $\mathcal{C}_{++}(\mathcal{X}) \cup \{0\}$ (see Lemma B.2 (iii)) and that $\{0\}$ and $\mathcal{C}_{++}(\mathcal{X})$ are two parts of $\mathcal{C}_+(\mathcal{X})$ with respect to the relation $\sim$ (see [32, Lemma 1.2.2]). Now, if $\mathsf{L}_\alpha f = \mathsf{L}_\alpha f' = 0$, then we have $d_H(\mathsf{L}_\alpha f, \mathsf{L}_\alpha f') = d_H(0,0) = 0$. Therefore it is sufficient to study the case that $\mathsf{L}_\alpha f, \mathsf{L}_\alpha f' \in \mathcal{C}_{++}(\mathcal{X})$. Following the characterization of Hilbert's metric on $\mathcal{C}_{++}(\mathcal{X})$ given in Lemma B.1, we have

$$
\begin{aligned}
d_H(\mathsf{L}_\alpha f, \mathsf{L}_\alpha f') &= \log\ \max_{x,y\in\mathcal{X}} \frac{(\mathsf{L}_\alpha f)(x)\,(\mathsf{L}_\alpha f')(y)}{(\mathsf{L}_\alpha f)(y)\,(\mathsf{L}_\alpha f')(x)} \\
&= \log\ \max_{x,y\in\mathcal{X}} \frac{\int k(x,z) f(z)\, d\alpha(z)\ \int k(y,w) f'(w)\, d\alpha(w)}{\int k(y,z) f(z)\, d\alpha(z)\ \int k(x,w) f'(w)\, d\alpha(w)} \\
&= \log\ \max_{x,y\in\mathcal{X}} \frac{\int k(x,z) k(y,w)\, f(z) f'(w)\, d\alpha(z) d\alpha(w)}{\int k(y,z) k(x,w)\, f(z) f'(w)\, d\alpha(z) d\alpha(w)} \\
&= \log\ \max_{x,y\in\mathcal{X}} \frac{\int \frac{k(x,z) k(y,w)}{k(y,z) k(x,w)}\, k(y,z) k(x,w)\, f(z) f'(w)\, d\alpha(z) d\alpha(w)}{\int k(y,z) k(x,w)\, f(z) f'(w)\, d\alpha(z) d\alpha(w)} \\
&\leq \log\ \max_{x,y,z,w\in\mathcal{X}} \frac{k(x,z) k(y,w)}{k(y,z) k(x,w)}.
\end{aligned}
$$

Since, for every $x,y \in \mathcal{X}$, $\mathsf{c}(x,y) \in [0, \mathsf{D}]$, we have $k(x,y) \in [e^{-\mathsf{D}/\varepsilon}, 1]$ and hence

$$d_H(\mathsf{L}_\alpha f, \mathsf{L}_\alpha f') \leq 2\mathsf{D}/\varepsilon. \qquad\square$$

A consequence of Proposition B.3 is a special case of Birkhoff-Hopf theorem.

**Theorem B.4** (Birkhoff-Hopf Theorem). *Let $\lambda = \frac{e^{\mathsf{D}/\varepsilon}-1}{e^{\mathsf{D}/\varepsilon}+1}$ and $\alpha \in \mathcal{M}_1^+(\mathcal{X})$. Then, for every $f, f' \in \mathcal{C}_+(\mathcal{X})$ such that $f \sim f'$, we have*

$$d_H(\mathsf{L}_\alpha f, \mathsf{L}_\alpha f') \leq \lambda\, d_H(f, f'). \tag{B.18}$$

*Proof.* The statement is a direct application of the Birkhoff-Hopf theory [32, Sections A.4 and A.7] The *Birkhoff contraction ratio* of $\mathsf{L}_\alpha$ is defined as

$$\kappa(\mathsf{L}_\alpha) = \inf\left\{\hat{\lambda} \in \mathbb{R}_+ \mid d_H(\mathsf{L}_\alpha f, \mathsf{L}_\alpha f') \leq \hat{\lambda} d_H(f, f')\ \forall f, f' \in \mathcal{C}_+(\mathcal{X}),\ f \sim f'\right\}.$$

Then it follows from Birkhoff-Hopf theorem [32, Theorem A.4.1] that

$$\kappa(\mathsf{L}_\alpha) = \tanh\left(\frac{1}{4}\Delta(\mathsf{L}_\alpha)\right). \tag{B.19}$$

Recalling the upper bound on the projective diameter f $\mathsf{L}_\alpha$ given in Proposition B.3, we have

$$\kappa(\mathsf{L}_\alpha) \leq \tanh\left(\frac{\mathsf{D}}{2\varepsilon}\right) = \frac{e^{\mathsf{D}/\varepsilon}-1}{e^{\mathsf{D}/\varepsilon}+1} = \lambda,$$

and (B.18) follows. $\qquad\square$

## B.2 DAD problems

**The map $A_\alpha$.** Let $\alpha \in \mathcal{M}_+^1(\mathcal{X})$. We define the map $A_\alpha \colon \mathcal{C}_{++}(\mathcal{X}) \to \mathcal{C}_{++}(\mathcal{X})$, such that

$$(\forall f \in \mathcal{C}_{++}(\mathcal{X})) \qquad A_\alpha(f) = R \circ L_\alpha(f) = 1/(L_\alpha f), \tag{B.20}$$

where $R \colon \mathcal{C}_{++}(\mathcal{X}) \to \mathcal{C}_{++}(\mathcal{X})$ is defined by $R(f) = 1/f$ with

$$(1/f) \colon x \mapsto \frac{1}{f(x)}. \tag{B.21}$$

Note that $A_\alpha$ is well defined since, by Lemma B.2 (iii), $L_\alpha(\mathcal{C}_{++}(\mathcal{X})) \subset \mathcal{C}_{++}(\mathcal{X})$ and, for every $f \in \mathcal{C}_{++}(\mathcal{X})$, $\min_\mathcal{X} f > 0$, being $\mathcal{X}$ compact. Moreover, it follows from (B.11) in Lemma B.1, that, for any two $f, f' \in \mathcal{C}_{++}(\mathcal{X})$

$$d_H(1/f, 1/f') = \log \max_{x,y \in \mathcal{X}} \frac{f(y)f'(x)}{f(x)f'(y)} = d_H(f, f'). \tag{B.22}$$

We highlight here the connection between $T_\alpha$ introduced in the main text in (3) and $A_\alpha$, namely for any $\alpha \in \mathcal{M}_1^+(\mathcal{X})$ and $u \in \mathcal{C}(\mathcal{X})$

$$T_\alpha(u) = \varepsilon \log(A_\alpha(e^{u/\varepsilon})). \tag{B.23}$$

**Dual $\mathrm{OT}_\varepsilon$ Problem.** We focus on the dual problem (2) of the optimal transport problem with entropic regularization. Let $\alpha, \beta \in \mathcal{M}_1^+(\mathcal{X})$ and $\varepsilon > 0$, we consider

$$\max_{u,v \in \mathcal{C}(\mathcal{X})} \int u(x)\,d\alpha + \int v(y)\,d\beta(y) - \varepsilon \int e^{\frac{u(x)+v(y)-c(x,y)}{\varepsilon}} \, d\alpha(x)d\beta(y). \tag{B.24}$$

The optimality conditions for problem (B.24) are

$$\begin{cases} e^{-\frac{u(x)}{\varepsilon}} = \displaystyle\int_\mathcal{X} e^{\frac{v(y)-c(x,y)}{\varepsilon}} \, d\beta(y) & (\forall\, x \in \mathrm{supp}(\alpha)) \\[2mm] e^{-\frac{v(y)}{\varepsilon}} = \displaystyle\int_\mathcal{X} e^{\frac{u(x)-c(x,y)}{\varepsilon}} \, d\alpha(x) & (\forall\, y \in \mathrm{supp}(\beta)), \end{cases} \tag{B.25}$$

which are equivalent to

$$\begin{cases} g(y)^{-1} = \displaystyle\int_\mathcal{X} e^{\frac{-c(x,y)}{\varepsilon}} f(x) \, d\alpha(x) & (\forall\, y \in \mathrm{supp}(\beta)) \\[2mm] f(x)^{-1} = \displaystyle\int_\mathcal{X} e^{\frac{-c(x,y)}{\varepsilon}} g(y) \, d\beta(y) & (\forall\, x \in \mathrm{supp}(\alpha)), \end{cases} \tag{B.26}$$

where $f = e^{u/\varepsilon} \in \mathcal{C}_{++}(\mathcal{X})$ and $g = e^{v/\varepsilon} \in \mathcal{C}_{++}(\mathcal{X})$. In the rest of the section we will consider the following *DAD problem* [32, 37]: find $f, g \in \mathcal{C}_{++}(\mathcal{X})$ such that

$$(\forall\, y \in \mathcal{X}) \ \int_\mathcal{X} f(x)k(x,y)g(y)\,d\alpha(x) = 1 \ \text{ and } \ (\forall\, x \in \mathcal{X}) \ \int_\mathcal{X} f(x)k(x,y)g(y)\,d\beta(y) = 1, \tag{B.27}$$

where $k$ is defined in (B.1). It is clear that a solution of (B.27) is also a solution of (B.26). However, the vice versa is in general not true, even though there is a canonical way to build solutions of (B.27) starting from solutions of (B.26): indeed if $(f, g)$ is a solution of (B.26), then the functions $\bar{f}, \bar{g} \colon \mathcal{X} \to \mathbb{R}$ defined through $\bar{f}(x)^{-1} = \int_\mathcal{X} k(x,y)g(y)\,d\beta(y)$ and $\bar{g}(y)^{-1} = \int_\mathcal{X} k(x,y)f(x)\,d\alpha(x)$ provide a solution of (B.27). So, the dual $\mathrm{OT}_\varepsilon$ problem (B.24) admits a solution if and only if the corresponding DAD problem (B.27) admits a solution. Recalling the definition of $A_\alpha$ in (B.20), problem (B.27) can be more compactly written as

$$f = A_\beta(g) \qquad \text{and} \qquad g = A_\alpha(f), \tag{B.28}$$

or equivalently, by setting $A_{\beta\alpha} = A_\beta \circ A_\alpha$ and $A_{\alpha\beta} = A_\alpha \circ A_\beta$,

$$f = A_{\beta\alpha}(f) \qquad \text{and} \qquad g = A_{\alpha\beta}(g). \tag{B.29}$$

This shows that the solutions of the DAD problem (B.27) are the fixed points of $\mathsf{A}_{\alpha\beta}$ and $\mathsf{A}_{\beta\alpha}$ respectively. Note that the operators $\mathsf{A}_{\beta\alpha}$ and $\mathsf{A}_{\alpha\beta}$ are positively homogeneous, that is, for every $t \in \mathbb{R}_{++}$ and $f \in \mathcal{C}_{++}(\mathcal{X})$, $\mathsf{A}_{\beta\alpha}(tf) = t\mathsf{A}_{\beta\alpha}(f)$ and $\mathsf{A}_{\alpha\beta}(tf) = t\mathsf{A}_{\alpha\beta}(f)$. Thus, if $f$ is a fixed point of $\mathsf{A}_{\beta\alpha}$, then $tf$ is also a fixed point of $\mathsf{A}_{\beta\alpha}$, for every $t > 0$. If $(f, g)$ is a solution of the DAD problem (B.27), then the pair $(u, v)$, with $u = \varepsilon \log f$ and $v = \varepsilon \log g$ is a solution of (B.24). We refer to these solutions as *Sinkhorn potentials* of the pair $(\alpha, \beta)$. Finally, note that, it follows from (B.25) that solutions of (B.24) are determined $(\alpha, \beta)$-a.e. on $\mathcal{X}$ and up to a translation of the form $(u + t, v - t)$, for some $t \in \mathbb{R}$.

The following result is essentially the specialization of [32, Thm. 7.1.4] to the case of the map $\mathsf{A}_{\beta\alpha}$. We report the proof here for completeness and the reader's convenience.

**Theorem B.5** (Hilbert's metric contraction for $\mathsf{A}_{\beta\alpha}$). *The map $\mathsf{A}_{\beta\alpha} : \mathcal{C}_{++}(\mathcal{X}) \to \mathcal{C}_{++}(\mathcal{X})$ has a unique fixed point up to positive scalar multiples. Moreover, let $\lambda = \frac{e^{\mathsf{D}/\varepsilon}-1}{e^{\mathsf{D}/\varepsilon}+1}$. Then, for every $f, f' \in \mathcal{C}_{++}(\mathcal{X})$,*

$$d_H(\mathsf{A}_{\beta\alpha}(f), \mathsf{A}_{\beta\alpha}(f')) \le \lambda^2 \, d_H(f, f'). \tag{B.30}$$

*Proof.* By combining (B.22) with Theorem B.4 we obtain that, for any $f, f' \in \mathcal{C}_{++}(\mathcal{X})$

$$d_H(\mathsf{A}_\alpha(f), \mathsf{A}_\alpha(f')) = d_H(1/(\mathsf{L}_\alpha f), 1/(\mathsf{L}_\alpha f')) = d_H(\mathsf{L}_\alpha f, \mathsf{L}_\alpha f') \le \lambda \, d_H(f, f'). \tag{B.31}$$

Since the same holds for $\mathsf{A}_\beta$ then (B.30) is satisfied. Now, let $C = \mathcal{C}_{++}(\mathcal{X}) \cap \partial B_1(0)$. Let $\overline{\mathsf{A}}_{\beta\alpha} : C \to C$ be the map such that

$$(\forall f \in C) \qquad \overline{\mathsf{A}}_{\beta\alpha}(f) = \frac{\mathsf{A}_{\beta\alpha}(f)}{\|\mathsf{A}_{\beta\alpha}(f)\|_\infty}. \tag{B.32}$$

Then, since $d_H(sf, tf') = d_H(f, f')$ for any $s, t > 0$ and $f, f' \in C$, we have

$$d_H(\overline{\mathsf{A}}_{\beta\alpha}(f), \overline{\mathsf{A}}_{\beta\alpha}(f')) = d_H(\mathsf{A}_{\beta\alpha}(f), \mathsf{A}_{\beta\alpha}(f')) \le \lambda^2 \, d_H(f, f'). \tag{B.33}$$

Since $(C, d_H)$ is a complete metric space [36, Theorem 1.2] and $\overline{\mathsf{A}}_{\beta\alpha}$ is a contraction, we can apply Banach's contraction theorem and conclude that there exists a unique fixed point of $\overline{\mathsf{A}}_{\beta\alpha}$, namely a function $\bar{f} \in C$ such that

$$\bar{f} = \overline{\mathsf{A}}_{\beta\alpha}(\bar{f}) = \frac{\mathsf{A}_{\beta\alpha}(\bar{f})}{\|\mathsf{A}_{\beta\alpha}(\bar{f})\|_\infty}. \tag{B.34}$$

Hence $\bar{f}$ is an eigenvector for $\mathsf{A}_{\beta\alpha}$ with eigenvalue $t = \|\mathsf{A}_{\beta\alpha}(\bar{f})\|_\infty > 0$. Now, we note that

$$(\forall \, f, g \in \mathcal{C}_{++}(\mathcal{X})) \quad \langle g\mathsf{L}_\alpha f, \beta \rangle = \langle f\mathsf{L}_\beta g, \alpha \rangle = \int_{\mathcal{X} \times \mathcal{X}} f(x)k(x,y)g(y)d(\alpha \otimes \beta)(x,y). \tag{B.35}$$

Set $\bar{g} = \mathsf{A}_\alpha(\bar{f})$, so that $\mathsf{A}_\beta(\bar{g}) = t\bar{f}$. Then, recalling the definitions of $\mathsf{A}_\alpha$ and $\mathsf{A}_\beta$, we have $\bar{g}\mathsf{L}_\alpha\bar{f} \equiv 1$ and $t^{-1} \equiv \bar{f}\mathsf{L}_\beta\bar{g}$. Hence $t^{-1} = \langle \bar{f}\mathsf{L}_\beta\bar{g}, \alpha \rangle = \langle \bar{g}\mathsf{L}_\alpha\bar{f}, \beta \rangle = 1$. Therefore $\bar{f}$ is a fixed point of $\mathsf{A}_{\beta\alpha}$. Finally, if $\bar{f}' \in \mathcal{C}_{++}(\mathcal{X})$ is a fixed point of $\mathsf{A}_{\beta\alpha}$, then, since $\mathsf{A}_{\beta\alpha}$ is positively homogeneous, we have

$$\overline{\mathsf{A}}_{\beta\alpha}(\bar{f}'/\|\bar{f}'\|_\infty) = \frac{\mathsf{A}_{\beta\alpha}(\bar{f}'/\|\bar{f}'\|_\infty)}{\|\mathsf{A}_{\beta\alpha}(\bar{f}'/\|\bar{f}'\|_\infty)\|_\infty} = \frac{\mathsf{A}_{\beta\alpha}(\bar{f}')}{\|\mathsf{A}_{\beta\alpha}(\bar{f}')\|_\infty} = \frac{\bar{f}'}{\|\bar{f}'\|_\infty}, \tag{B.36}$$

that is, $\bar{f}'/\|\bar{f}'\|_\infty$ is a fixed point of $\overline{\mathsf{A}}_{\beta\alpha}$. Thus, $\bar{f}'/\|\bar{f}'\|_\infty = \bar{f}$ and hence $\bar{f}'$ is a multiple of $\bar{f}$. $\quad\square$

**Corollary B.6** (Existence and uniqueness of Sinkhorn potentials). *Let $\alpha, \beta \in \mathcal{M}_+^1(\mathcal{X})$. Then, the DAD problem (B.27) admits a solution $(f, g)$ and every other solution is of type $(tf, t^{-1}g)$, for some $t > 0$. Moreover, there exists a pair $(u, v) \in \mathcal{C}(\mathcal{X})^2$ of Sinkhorn potentials and every other pair of Sinkhorn potentials is of type $(u + s, v - s)$, for some $s \in \mathbb{R}$. In particular, for every $x_o \in \mathcal{X}$, there exist a unique pair $(u, v)$ of Sinkhorn potentials such that $u(x_0) = 0$.*

*Proof.* It follows from Theorem B.5 and the discussion after (B.29). $\quad\square$

**Bounding $(f, g)$ point-wise.** We conclude this section by providing additional properties of the solutions $(f, g)$ of the DAD problem (B.28). In particular, we show that there exists one such solution for which it is possible to provide a point-wise upper and lower bound independent on $\alpha$ and $\beta$.

**Remark B.7.** *Let $f \in \mathcal{C}_{++}(\mathcal{X})$ and set $g = \mathsf{A}_\alpha(f)$. Then, recalling (B.20) and (B.4), we have that, for every $x \in \mathcal{X}$,*

$$1 = g(x)(\mathsf{L}_\alpha\, f)(x) \leq g(x)\,\|\mathsf{L}_\alpha f\|_\infty \leq g(x)\,\|f\|_\infty$$

*and*

$$1 = g(x)(\mathsf{L}_\alpha\, f)(x) \geq g(x)(\min_{\mathcal{X}} f) \int \mathsf{k}(x,z)\, d\alpha(z) \geq g(x)(\min_{\mathcal{X}} f) e^{-\mathsf{D}/\varepsilon}.$$

*Therefore,*

$$\min_{\mathcal{X}} g \geq \frac{1}{\|f\|_\infty} \quad \text{and} \quad \|g\|_\infty \leq \frac{e^{\mathsf{D}/\varepsilon}}{\min_{\mathcal{X}} f}. \tag{B.37}$$

**Lemma B.8.** *(Auxiliary Cone) Consider the set*

$$K = \{f \in \mathcal{C}_+(\mathcal{X}) \mid f(x) \leq f(y)\, e^{\mathsf{D}/\varepsilon} \ \forall x, y \in \mathcal{X}\}. \tag{B.38}$$

*Let $\alpha \in \mathcal{M}_+^1(\mathcal{X})$. Then the following holds.*

    (i) *$K$ is a closed convex cone and $K \subset \mathcal{C}_{++}(\mathcal{X}) \cup \{0\}$;*

    (ii) *$\mathsf{L}_\alpha(\mathcal{C}_+(\mathcal{X})) \subset K$;*

    (iii) *$\mathsf{R}(K) \subset K$;*

    (iv) *$\mathrm{Ran}(\mathsf{A}_\alpha) \subset K$;*

    (v) *If $f \in K$ and $g = \mathsf{A}_\alpha f$, then $g \in K$ and $1 \leq (\min_{\mathcal{X}} g)\,\|f\|_\infty \leq \|g\|_\infty\,\|f\|_\infty \leq e^{2\mathsf{D}/\varepsilon}$.*

    (vi) *If $f \in K$ is such that $f(x_o) = 1$ for some $x_o \in \mathcal{X}$, then $\|\varepsilon \log f\|_\infty \leq \mathsf{D}$.*

*Proof.* (i): We see that for any $f \in K$,

$$\max_{\mathcal{X}} f \leq (\min_{\mathcal{X}} f)\, e^{\mathsf{D}/\varepsilon}, \tag{B.39}$$

so, if $f(x) = 0$ for some $x \in \mathcal{X}$, then $f(x) = 0$ on all $\mathcal{X}$. Hence $K \subseteq \mathcal{C}_{++}(\mathcal{X}) \cup \{0\}$. It is straightforward to verify that $K$ is a convex cone. Moreover $K$ is also closed. Indeed if $(f_n)_{n \in \mathbb{N}}$ is a sequence in $K$ which converges uniformly to $f \in \mathcal{C}(\mathcal{X})$, then, for every $x, y \in \mathcal{X}$ and every $n \in \mathbb{N}$, $f_n(x) \leq f_n(y)e^{\mathsf{D}/\varepsilon}$ and hence, letting $n \to +\infty$, we have $f(x) \leq f(y)e^{\mathsf{D}/\varepsilon}$.

(ii): For every $f \in \mathcal{C}_+(\mathcal{X})$ and $x, y \in \mathcal{X}$, we have

$$\begin{aligned}
(\mathsf{L}_\alpha f)(x) &= \int \mathsf{k}(x, z) f(z)\, d\alpha(z) \\
&= \int \frac{\mathsf{k}(x, z)}{\mathsf{k}(y, z)}\, \mathsf{k}(y, z) f(z)\, d\alpha(z) \\
&\leq e^{\mathsf{D}/\varepsilon} \int \mathsf{k}(y, z) f(z)\, d\alpha(z) \\
&= e^{\mathsf{D}/\varepsilon} (\mathsf{L}_\alpha f)(y).
\end{aligned}$$

(iii): For every $f \in K$,

$$(\forall\, x, y \in \mathcal{X}) \qquad f(x) \leq f(y)\, e^{\mathsf{D}/\varepsilon} \iff \frac{1}{f(y)} \leq \frac{1}{f(x)}\, e^{\mathsf{D}/\varepsilon}.$$

(iv) It follows from (ii) and (iii) and the definitions of $\mathsf{A}_\alpha$.

(v): It follows from (iv), (B.37), and (B.39).

(vi): Let $f \in K$ be such that $f(x_o) = 1$. Then $\min_{\mathcal{X}} f \leq 1 \leq \max_{\mathcal{X}} f$. Thus, it follows from (B.39) that

$$\max_{\mathcal{X}} f \leq e^{\mathsf{D}/\varepsilon} \quad \text{and} \quad \min_{\mathcal{X}} f \geq e^{-\mathsf{D}/\varepsilon} \tag{B.40}$$

and hence, for every $x \in \mathcal{X}$, $-\mathsf{D} \leq \varepsilon \log f(x) \leq \mathsf{D}$. $\qquad\square$

As a direct consequence of Lemma B.8 we can establish a uniform point-wise upper and lower bound for the value of DAD solutions.

**Corollary B.9.** *Let $\alpha, \beta \in \mathcal{M}_1^+(\mathcal{X})$. Let $x_o \in \mathcal{X}$ and let $(f, g)$ be the solution of (B.28) such that $f(x_o) = 1$. Then $\|f\|_\infty \leq e^{\mathsf{D}/\varepsilon}$ and $\|g\|_\infty \leq e^{2\mathsf{D}/\varepsilon}$. Moreover, the corrisponding pair $(u, v)$ of Sinkhorn potentials satifies $\|u\|_\infty \leq \mathsf{D}$ and $\|v\|_\infty \leq 2\mathsf{D}$.*

*Proof.* Since $f$ and $g$ are fixed points of $\mathsf{A}_{\beta\alpha}$ and $\mathsf{A}_{\alpha\beta}$ respectively, it follows from Lemma B.8 (iv) that $f, g \in K$. Then, Lemma B.8 (vi) yields $\|f\|_\infty \leq e^{\mathsf{D}/\varepsilon}$, whereas by the second of (B.37) and (B.40) we derive that $\|g\|_\infty \leq e^{2\mathsf{D}/\varepsilon}$. $\qquad\qquad\square$

## B.3 Sinkhorn-Knopp algorithm in infinite dimension

In the context of optimal transport, Sinkhorn-Knopp algorithm is often presented and studied in finite dimension [13, 38]. The algorithm originates from so called *matrix scaling problems*, also called *DAD problems*, which consists in finding, for a given matrix $A$ with nonnegative entries, two diagonal matrices $D_1$, $D_2$ such that $D_1 A D_2$ is doubly stochastic [41]. In our setting it is crucial to analyze the algorithm in infinite dimension.

Theorem B.5 shows that $\mathsf{A}_{\beta\alpha}$ is a contraction with respect to the Hilbert's metric. This suggests a direct approach to find the solutions of the DAD problem by adopting a fixed-point strategy, which turns out to applying the operators $\mathsf{A}_\alpha$ and $\mathsf{A}_\beta$ alternatively, starting from some $f^{(0)} \in \mathcal{C}_{++}(\mathcal{X})$. This is exactly the approach to the Sinkhorn algorithm pioneered by [22, 34] and further developed in an infinite dimensional setting in [37]. In this section we review the algorithm and give the convergence properties for the special kernel k in (B.1). In particular we provide rate of convergence in the sup norm $\|\cdot\|_\infty$.

---

**Algorithm B.1** Sinkhorn-Knopp algorithm (infinite dimensional case)

Let $\alpha, \beta \in \mathcal{M}_+^1(\mathcal{X})$. Let $f^{(0)} \in \mathcal{C}_{++}(\mathcal{X})$ and define,

$$
\text{for } \ell = 0, 1, \dots
$$
$$
\left|\begin{array}{l} g^{(\ell+1)} = \mathsf{A}_\alpha(f^{(\ell)}) \\ f^{(\ell+1)} = \mathsf{A}_\beta(g^{(\ell+1)}) \end{array}\right.
$$

---

**Theorem B.10** (Convergence of Sinkhorn-Knopp algorithm). *Let $(f^{(\ell)})_{\ell \in \mathbb{N}}$ be defined according to Algorithm B.1. Let $x_o \in \mathcal{X}$ and let $(f, g)$ be the solution of the DAD problem (B.26) such that $f(x_o) = 1$. Then, defining $\lambda$ according to Theorem B.5 and, for every $\ell \in \mathbb{N}$, $\tilde{f}^{(\ell)} = f^{(\ell)}/f^{(\ell)}(x_o)$ and $\tilde{g}^{(\ell+1)} = g^{(\ell+1)} f^{(\ell)}(x_o)$, we have*

$$
\begin{cases} \|\log \tilde{f}^{(\ell)} - \log f\|_\infty \leq \lambda^{2\ell}\left(\dfrac{\mathsf{D}}{\varepsilon} + \log \dfrac{\|f^{(0)}\|_\infty}{\min_{\mathcal{X}} f^{(0)}}\right) \\[4mm] \|\log \tilde{g}^{(\ell+1)} - \log g\|_\infty \leq e^{3\mathsf{D}/\varepsilon}\|\log \tilde{f}^{(\ell)} - \log f\|_\infty. \end{cases}
\tag{B.41}
$$

*Moreover, let the potentials $(u, v) = (\varepsilon \log f, \varepsilon \log g)$ and, for every $\ell \in \mathbb{N}$, $(\tilde{u}^{(\ell)}, \tilde{v}^{(\ell)}) = (\varepsilon \log \tilde{f}^{(\ell)}, \varepsilon \log \tilde{g}^{(\ell)})$. Then we have*

$$
\|\tilde{u}^{(\ell)} - u\|_\infty \leq \lambda^{2\ell}\left(\frac{\mathsf{D} + \max_{\mathcal{X}} u^{(0)} - \min_{\mathcal{X}} u^{(0)}}{\varepsilon}\right).
\tag{B.42}
$$

*Proof.* Let $\mathcal{A}$ be the set in Lemma C.1. Clearly, for every $\ell \in \mathbb{N}$, we have $f^{(\ell+1)} = \mathsf{A}_{\beta\alpha}(f^{(\ell)})$ and $\bar{f}, \tilde{f}^\ell \in \mathcal{A}$. Thus, it follows from Theorem B.5 and (C.2) in Lemma C.1 that, for every $\ell \in \mathbb{N}$,

$$
\|\log \tilde{f}^{(\ell)} - \log f\|_\infty \leq d_H(\tilde{f}^\ell, f) = d_H(\mathsf{A}_{\beta\alpha}^{(\ell)}(f^{(0)}), f) \leq \lambda^{2\ell} d_H(f^{(0)}, f).
$$

Moreover, recalling (B.11), we have

$$
d_H(f^{(0)}, f) = d_H(1/f^{(0)}, \mathsf{L}_\beta g) = \log \max_{x,y \in \mathcal{X}} \frac{f^{(0)}(y)\mathsf{L}_\beta g(y)}{f^{(0)}(x)\mathsf{L}_\beta g(x)} \leq \log\left[e^{\mathsf{D}/\varepsilon} \max_{x,y \in \mathcal{X}} \frac{f^{(0)}(y)}{f^{(0)}(x)}\right]
$$

where we used the fact that $L_\beta(\mathcal{C}_{++}(\mathcal{X})) \subset K$ and the definition (B.38). Thus, the first inequality in (B.41) follows. The second inequality in (B.41) and (B.42) follow directly from Lemma C.3 and the fact that $u^{(0)} = \varepsilon \log f^{(0)}$. □

---

**Algorithm B.2** Sinkhorn-Knopp algorithm (finite dimensional case)

---

Let $\mathsf{M} \in \mathbb{R}_{++}^{n_1 \times n_2}$, $\mathsf{a} \in \mathbb{R}_+^{n_1}$, with $\mathsf{a}^\top 1_{n_1} = 1$, and $\mathsf{b} \in \mathbb{R}_+^{n_2}$, with $\mathsf{b}^\top 1_{n_2} = 1$. Let $\mathsf{f}^{(0)} \in \mathbb{R}_{++}^{n_1}$ and define

$$\text{for } \ell = 0, 1, \dots$$

$$\left| \begin{aligned} \mathsf{g}^{(\ell+1)} &= \frac{\mathsf{b}}{\mathsf{M}^\top \mathsf{f}^{(\ell)}} \\ \mathsf{f}^{(\ell+1)} &= \frac{\mathsf{a}}{\mathsf{M}\mathsf{g}^{(\ell+1)}}. \end{aligned} \right.$$

---

**Proposition B.11.** *Suppose that $\alpha$ and $\beta$ are probability measures with finite support. Then Algorithm B.1 can be reduced to the finite dimensional Algorithm B.2. More specifically, suppose that $\alpha = \sum_{i=1}^{n_1} a_i \delta_{x_i}$, and $\beta = \sum_{i=1}^{n_2} b_i \delta_{y_i}$, where $\mathsf{a} = (a_i)_{1 \le i \le n_1} \in \mathbb{R}_+^{n_1}$, $\sum_{i=1}^n a_i = 1$ and $\mathsf{b} = (b_i)_{1 \le i \le n_2} \in \mathbb{R}_+^{n_2}$, $\sum_{i=1}^n b_i = 1$. Let $\mathsf{K} \in \mathbb{R}^{n_1 \times n_2}$ be such that $\mathsf{K}_{i_1, i_2} = \mathsf{k}(x_{i_1}, y_{i_2})$ and let $\mathsf{M} = \mathrm{diag}(\mathsf{a})\mathsf{K}\mathrm{diag}(\mathsf{b}) \in \mathbb{R}^{n_1 \times n_2}$. Let $(\mathsf{f}^{(\ell)})_{\ell \in \mathbb{N}}$ and $(f^{(\ell)})_{\ell \in \mathbb{N}}$ be defined according to Algorithm B.2 and Algorithm B.1 respectively, with $\mathsf{f}^{(0)} = (f^{(0)}(x_i))_{1 \le i \le n_1}$. Then, for every $\ell \in \mathbb{N}$,*

$$(\forall x \in \mathcal{X})(\forall y \in \mathcal{X}) \, g^{(\ell+1)}(y)^{-1} = \sum_{i_1=1}^{n_1} k(x_{i_1}, y)a_{i_1}\mathsf{f}_{i_1}^{(\ell)} \text{ and } f^{(\ell+1)}(x)^{-1} = \sum_{i_2=1}^{n_2} k(x, y_{i_2})b_{i_2}\mathsf{g}_{i_2}^{(\ell+1)}.$$

*Moreover, setting $u^{(\ell)} = \varepsilon \log f^{(\ell)}$, $v^{(\ell)} = \varepsilon \log g^{(\ell)}$, $\mathsf{u}^{(\ell)} = \varepsilon \log \mathsf{f}^{(\ell)}$, and $\mathsf{v}^{(\ell)} = \varepsilon \log \mathsf{g}^{(\ell)}$, we have*

$$\left\{ \begin{aligned} (\forall y \in \mathcal{X}) \quad v^{(\ell+1)}(y) &= -\varepsilon \log \sum_{i_1=1}^{n_1} \exp(\mathsf{u}_{i_1}^{(\ell)} - \mathsf{c}(x_{i_1}, y))a_{i_1} \\ (\forall x \in \mathcal{X}) \quad u^{(\ell+1)}(x) &= -\varepsilon \log \sum_{i_2=1}^{n_2} \exp(\mathsf{v}_{i_2}^{(\ell+1)} - \mathsf{c}(x, y_{i_2}))b_{i_2}. \end{aligned} \right. \tag{B.43}$$

*Proof.* Since $\alpha$ and $\beta$ have finite support, we derive from the definitions of $f^{(\ell+1)}$ and $g^{(\ell+1)}$ in Algorithm B.1 and that of $\mathsf{A}_\alpha$ and $\mathsf{A}_\beta$ that

$$\left\{ \begin{aligned} (\forall x \in \mathcal{X}) \quad g^{(\ell+1)}(y)^{-1} &= (\mathsf{L}_\alpha f^{(\ell)})(y) = \sum_{i_1=1}^{n_1} a_{i_1}\mathsf{k}(x_{i_1}, y)f^{(\ell)}(x_{i_1}) \\ (\forall y \in \mathcal{X}) \quad f^{(\ell+1)}(x)^{-1} &= (\mathsf{L}_\beta g^{(\ell+1)})(x) = \sum_{i_2=1}^{n_2} \mathsf{k}(x, y_{i_2})b_{i_2}g^{(\ell+1)}(y_{i_2}). \end{aligned} \right.$$

Now, multiplying the above equations by $b_{i_2}$ and $a_{i_1}$ respectively, and recalling that $\mathsf{M}_{i_1, i_2} = a_{i_1}\mathsf{k}(x_{i_1}, y_{i_2})b_{i_2}$, we have

$$\begin{bmatrix} b_1 g^{(\ell+1)}(y_1)^{-1} \\ \vdots \\ b_{n_2}g^{(\ell+1)}(y_{n_2})^{-1} \end{bmatrix} = \mathsf{M}^\top \begin{bmatrix} f^{(\ell)}(x_1) \\ \vdots \\ f^{(\ell)}(x_{n_1}) \end{bmatrix}, \quad \begin{bmatrix} a_1 f^{(\ell+1)}(x_1)^{-1} \\ \vdots \\ a_{n_1}f^{(\ell+1)}(x_{n_1})^{-1} \end{bmatrix} = \mathsf{M} \begin{bmatrix} g^{(\ell+1)}(y_1) \\ \vdots \\ g^{(\ell+1)}(y_{n_2}) \end{bmatrix},$$

and hence

$$\begin{bmatrix} g^{(\ell+1)}(y_1) \\ \vdots \\ g^{(\ell+1)}(y_{n_2}) \end{bmatrix} = \mathsf{b} \Big/ \mathsf{M}^\top \begin{bmatrix} f^{(\ell)}(x_1) \\ \vdots \\ f^{(\ell)}(x_{n_1}) \end{bmatrix}, \quad \begin{bmatrix} f^{(\ell+1)}(x_1) \\ \vdots \\ f^{(\ell+1)}(x_{n_1}) \end{bmatrix} = \mathsf{a} \Big/ \mathsf{M} \begin{bmatrix} g^{(\ell+1)}(y_1) \\ \vdots \\ g^{(\ell+1)}(y_{n_2}) \end{bmatrix}.$$

Therefore, since $\mathsf{f}^{(0)} = (f^{(0)}(x_i))_{1 \le i \le n_1}$, recalling Algorithm B.2, it follows by induction that, for every $\ell \in \mathbb{N}$, $\mathsf{f}^{(\ell)} = (f^{(\ell)}(x_i))_{1 \le i \le n_1}$ and $\mathsf{g}^{(\ell)} = (g^{(\ell)}(x_i))_{1 \le i \le n_1}$. Thus, the first part of the statement follows. The second part follows directly from the definitions of $u^{(\ell)}$, $v^{(\ell)}$, $\mathsf{u}^{(\ell)}$, and $\mathsf{v}^{(\ell)}$. □

**Remark B.12.**

(i) Algorithm B.2 is the classical (discrete) Sinkhorn algorithm which was recently studied in several papers [13]. It follows from Theorem B.10 that considering the solution $(f, g)$ of the DAD problem such that $f(x_1) = 1$ and defining $\tilde{\mathsf{f}}^{(\ell)} = \mathsf{f}^{(\ell)}/\mathsf{f}_0^{(\ell)}$ and $\tilde{\mathsf{g}}^{(\ell)} = \mathsf{g}^{(\ell)}\mathsf{f}_0^{(\ell)}$, and $\mathsf{f}_i = f(x_i)$ and $\mathsf{g}_j = g(y_j)$, we have

$$\|\log \tilde{\mathsf{f}}^{(\ell)} - \log \mathsf{f}\|_\infty \leq \lambda^{2\ell}\left(\frac{\mathsf{D}}{\varepsilon} + \log \frac{\max_i \mathsf{f}_i^{(0)}}{\min_i \mathsf{f}_i^{(0)}}\right).$$

(ii) The procedure SINKHORNKNOPP discussed in the paper and called in Algorithm 2, actually output the vector $\mathsf{v} = \varepsilon \log \mathsf{g}^{(\ell)}$ for sufficiently large $\ell$.

(iii) Referring to Section 4 in the paper, we recognize that the expressions on the right hand side of (B.43) are precisely $\mathsf{T}_\alpha(u^{(\ell)})(x)$ and $\mathsf{T}_\beta(v^{(\ell+1)})(x)$ respectively.

# C  Lipschitz continuity of the gradient of Sinkhorn divergence with respect to the Total Variation

In this section we show that the gradient of the Sinkhorn divergence is Lipschitz continuous with respect to the Total Variation on $\mathcal{M}_1^+(\mathcal{X})$.

We start by characterizing the relation between the Hilbert's metric and the metric induced by the norm $\|\cdot\|_\infty$.

**Lemma C.1.** *Let $f, f' \in \mathcal{C}_{++}(\mathcal{X})$ and set $u = \varepsilon \log f$ and $u' = \varepsilon \log f'$. Then*

$$d_H(f, f') \leq 2\|\log f - \log f'\|_\infty \quad \text{or, equivalently} \quad d_H(e^{u/\varepsilon}, e^{u'/\varepsilon}) \leq \frac{2}{\varepsilon}\|u - u'\|_\infty. \quad \text{(C.1)}$$

*Moreover, let $x_o \in \mathcal{X}$, consider the sets $\mathcal{A} = \{h \in \mathcal{C}_{++}(\mathcal{X}) \mid h(x_o) = 1\}$ and $\mathcal{B} = \{w \in \mathcal{C}(\mathcal{X}) \mid w(x_o) = 0\}$. Suppose that $f, f' \in \mathcal{A}$ (or equivalently that $u, u' \in \mathcal{B}$). Then*

$$\frac{1}{2}d_H(f, f') \leq \|\log f - \log f'\|_\infty \leq d_H(f, f'). \quad \text{(C.2)}$$

*and*

$$\frac{\varepsilon}{2}d_H(e^{u/\varepsilon}, e^{u'/\varepsilon}) \leq \|u - u'\|_\infty \leq \varepsilon\, d_H(e^{u/\varepsilon}, e^{u'/\varepsilon}). \quad \text{(C.3)}$$

*Proof.* We have

$$\begin{aligned}
d_H(f, f') &= \log \max_{x,y \in \mathcal{X}} \frac{f(x)f'(y)}{f(y)f'(x)} \\
&= \log \max_{x \in \mathcal{X}} \frac{f(x)}{f'(x)} + \log \max_{y \in \mathcal{X}} \frac{f'(y)}{f(y)} \\
&= \max_{x \in \mathcal{X}} \log \frac{f(x)}{f'(x)} + \max_{y \in \mathcal{X}} \log \frac{f'(y)}{f(y)} \\
&\leq 2 \max_{x \in \mathcal{X}} \left|\log \frac{f(x)}{f'(x)}\right| \\
&= 2\|\log(f/f')\|_\infty \\
&= 2\|\log f - \log f'\|_\infty
\end{aligned}$$

and (C.1) follows. Suppose that $f, f' \in \mathcal{A}$. Then

$$\begin{aligned}
\|\log f - \log f'\|_\infty &= \max\left\{\log \max_{x \in \mathcal{X}} \frac{f(x)}{f'(x)}, \log \max_{x \in \mathcal{X}} \frac{f'(x)}{f(x)}\right\} \\
&= \max\left\{\log \max_{x \in \mathcal{X}} \frac{f(x)f'(\bar{x})}{f(\bar{x})f'(x)}, \log \max_{x \in \mathcal{X}} \frac{f(\bar{x})f'(x)}{f(x)f'(\bar{x})}\right\} \\
&\leq \max\left\{\log \max_{x,y \in \mathcal{X}} \frac{f(x)f'(y)}{f(y)f'(x)}, \log \max_{x,y \in \mathcal{X}} \frac{f(y)f'(x)}{f(x)f'(y)}\right\} \\
&= d_H(f, f'),
\end{aligned}$$

since $f(x_o)/f'(x_o) = f'(x_o)/f(x_o) = 1$. Therefore, (C.2) follows. $\qquad\square$

**Lemma C.2.** *For every $x, y \in \mathbb{R}_{++}$ we have*

$$|\log x - \log y| \le \max\left\{x^{-1}, y^{-1}\right\}|x - y|. \tag{C.4}$$

The following result allows to extend the previous observations on a pair $f, f'$ to the corresponding $g = \mathsf{A}_\alpha f$ and $g' = \mathsf{A}_\alpha f'$.

**Lemma C.3.** *Let $x_o \in \mathcal{X}$ and $K \subset \mathcal{C}_+(\mathcal{X})$ the cone from Lemma B.8. Let $f, f' \in K$ be such that $f(x_o) = f'(x_o) = 1$, and set $g = \mathsf{A}_\alpha f$ and $g' = \mathsf{A}_\alpha f'$. Then,*

$$\|\log g - \log g'\|_\infty \le e^{3\mathsf{D}/\varepsilon} \|\log f - \log f'\|_\infty. \tag{C.5}$$

*Proof.* It follows from (B.20) and Lemma C.2 that

$$|\log g - \log g'| = \left|\log \frac{g}{g'}\right| = \left|\log \frac{\mathsf{L}_\alpha f'}{\mathsf{L}_\alpha f}\right| \le \max\{g', g\}|\mathsf{L}_\alpha f - \mathsf{L}_\alpha f'|.$$

Therefore, since $1 \le \|f\|_\infty, \|f'\|_\infty$, and recalling Lemma B.8 (v) and (B.4), we have

$$\begin{aligned}
\|\log g - \log g'\|_\infty &\le \max\{\|g\|_\infty, \|g'\|_\infty\} \|\mathsf{L}_\alpha f - \mathsf{L}_\alpha f'\|_\infty \\
&\le \max\{\|f\|_\infty \|g\|_\infty, \|f'\|_\infty \|g'\|_\infty\} \|\mathsf{L}_\alpha f - \mathsf{L}_\alpha f'\|_\infty \\
&\le e^{2\mathsf{D}/\varepsilon} \|f - f'\|_\infty \\
&= e^{2\mathsf{D}/\varepsilon}\|e^{\log f} - e^{\log f'}\|_\infty.
\end{aligned}$$

Now, since $f, f' \le e^{\mathsf{D}/\varepsilon}$, we have $\log f, \log f' \le \mathsf{D}/\varepsilon$. Thus, the statement follows by noting that the exponential function is Lipschitz continuous on $]-\infty, \mathsf{D}/\varepsilon]$ with constant $e^{\mathsf{D}/\varepsilon}$. $\qquad\square$

We are ready to prove the main result of the section.

**Theorem C.4** (Lipschitz continuity of the Sinkhorn potentials with respect to the total variation). *Let $\alpha, \beta, \alpha', \beta' \in \mathcal{M}_1^+(\mathcal{X})$ and let $x_o \in \mathcal{X}$. Let $(u, v), (u', v') \in \mathcal{C}(\mathcal{X})^2$ be the two pairs of Sinkhorn potentials corresponding to the solution of the regularized OT problem in (B.24) for $(\alpha, \beta)$ and $(\alpha', \beta')$ respectively such that $u(x_o) = u'(x_o) = 0$. Then*

$$\|u - u'\|_\infty \le 2\varepsilon e^{3\mathsf{D}/\varepsilon} \|(\alpha - \alpha', \beta - \beta')\|_{TV}. \tag{C.6}$$

*Hence, the map which, for each pair of probability distributions $(\alpha, \beta) \in \mathcal{M}_1^+(\mathcal{X})^2$ associates the component $u$ of the corresponding Sinkhorn potentials is $2\varepsilon e^{3\mathsf{D}/\varepsilon}$-Lipschitz continuous with respect to the total variation.*

*Proof.* The functions $f = e^{u/\varepsilon}$ and $f' = e^{u'/\varepsilon}$ are fixed points of the maps $\mathsf{A}_{\beta\alpha}$ and $\mathsf{A}_{\beta'\alpha'}$ respectively. Then, it follows from Theorem B.5 that

$$\begin{aligned}
d_H(f, f') &= d_H(\mathsf{A}_{\beta\alpha}(f), \mathsf{A}_{\beta'\alpha'}(f')) \\
&\le d_H(\mathsf{A}_{\beta\alpha}(f), \mathsf{A}_{\beta'\alpha'}(f)) + d_H(\mathsf{A}_{\beta'\alpha'}(f), \mathsf{A}_{\beta'\alpha'}(f')) \\
&\le d_H(\mathsf{A}_{\beta\alpha}(f), \mathsf{A}_{\beta'\alpha'}(f)) + \lambda^2 d_H(f, f'),
\end{aligned}$$

hence,

$$d_H(f, f') \le \frac{1}{1 - \lambda^2} d_H(\mathsf{A}_{\beta\alpha}(f), \mathsf{A}_{\beta'\alpha'}(f)). \tag{C.7}$$

Moreover, using (C.1), we have

$$\begin{aligned}
d_H(\mathsf{A}_{\beta\alpha}(f), \mathsf{A}_{\beta'\alpha'}(f)) &\le d_H(\mathsf{A}_{\beta\alpha}(f), \mathsf{A}_{\beta'\alpha}(f)) + d_H(\mathsf{A}_{\beta'\alpha}(f), \mathsf{A}_{\beta'\alpha'}(f)) \\
&\le d_H(\mathsf{A}_\beta(g), \mathsf{A}_{\beta'}(g)) + \lambda d_H(\mathsf{A}_\alpha(f), \mathsf{A}_{\alpha'}(f)) \\
&\le 2\left\|\log \frac{\mathsf{A}_\beta(g)}{\mathsf{A}_{\beta'}(g)}\right\|_\infty + 2\lambda \left\|\log \frac{\mathsf{A}_\alpha(f)}{\mathsf{A}_{\alpha'}(f)}\right\|_\infty. \tag{C.8}
\end{aligned}$$

Now, note that by Lemma C.2

$$\left| \log \frac{\mathsf{A}_\beta(g)}{\mathsf{A}_{\beta'}(g)} \right| = \left| \log \frac{\mathsf{L}_{\beta'}g}{\mathsf{L}_\beta g} \right| \le \max\{1/\mathsf{L}_\beta g, 1/\mathsf{L}_{\beta'}g\}|(\mathsf{L}_{\beta'} - \mathsf{L}_\beta)g| \tag{C.9}$$

and that, for every $x \in \mathcal{X}$,

$$\begin{aligned}
[(\mathsf{L}_{\beta'} - \mathsf{L}_\beta)g](x) &= \int \mathsf{k}(x,z)g(z)\,d(\beta - \beta')(z) \\
&= \langle \mathsf{k}(x,\cdot)g, \beta - \beta' \rangle \le \|g\|_\infty \|\beta - \beta'\|_{TV},
\end{aligned} \tag{C.10}$$

and, similarly, $[(\mathsf{L}_\beta - \mathsf{L}_{\beta'})g](x) \le \|g\|_\infty \|\beta - \beta'\|_{TV}$. Therefore, since $1/(\mathsf{L}_\beta g) = \mathsf{A}_\beta(g) = f$ and $\mathsf{L}_{\beta'}g \ge e^{-\mathsf{D}/\varepsilon} \min g$, it follows from Lemma B.8 (v) and (B.39) (applied to $g$) that

$$\left\| \log \frac{\mathsf{A}_\beta(g)}{\mathsf{A}_{\beta'}(g)} \right\|_\infty \le \max\left\{ \|f\|_\infty, \frac{e^{\mathsf{D}/\varepsilon}}{\min g} \right\} \|g\|_\infty \|\beta - \beta'\|_{TV} \le e^{2\mathsf{D}/\varepsilon} \|\beta - \beta'\|_{TV}. \tag{C.11}$$

Analogously, we have

$$\left\| \log \frac{\mathsf{A}_\alpha(f)}{\mathsf{A}_{\alpha'}(f)} \right\|_\infty \le e^{2\mathsf{D}/\varepsilon} \|\alpha - \alpha'\|_{TV}. \tag{C.12}$$

Putting (C.7), (C.8), (C.11), and (C.12) together, we have

$$d_H(f, f') \le \frac{2e^{2\mathsf{D}/\varepsilon}}{1 - \lambda^2} \left( \lambda \|\alpha - \alpha'\|_{TV} + \|\beta - \beta'\|_{TV} \right). \tag{C.13}$$

Now, note that since $e^{\mathsf{D}/\varepsilon} \ge 1$

$$\frac{1}{1 - \lambda^2} = \frac{(e^{\mathsf{D}/\varepsilon} + 1)^2}{4e^{\mathsf{D}/\varepsilon}} \le e^{\mathsf{D}/\varepsilon}. \tag{C.14}$$

Finally, recalling (C.3), we have

$$\|u - u'\|_\infty \le 2\varepsilon e^{3\mathsf{D}/\varepsilon} \|(\alpha - \alpha', \beta - \beta')\|_{TV}, \tag{C.15}$$

where $\|(\alpha - \alpha', \beta - \beta')\|_{TV} = \|\alpha - \alpha'\|_{TV} + \|\beta - \beta'\|_{TV}$ is the total variation norm on $\mathcal{M}(\mathcal{X})^2$. $\square$

**Corollary C.5.** *Under the assumption of Theorem C.4, we have*

$$\|u - u'\|_\infty + \|v - v'\|_\infty \le 2\varepsilon e^{3\mathsf{D}/\varepsilon}(1 + \varepsilon e^{3\mathsf{D}/\varepsilon}) \|(\alpha - \alpha', \beta - \beta')\|_{TV}. \tag{C.16}$$

*Proof.* It follows from Theorem C.4 and Lemma C.3. $\square$

We finally address the issue of the differentiability of the Sinkhorn divergence. We first recall a few facts about the directional differentiability of $\mathrm{OT}_\varepsilon$ briefly recalled in Section 2 of the main text. For a more in-depth analysis on this topic we refer to [21, Proposition 2]. See also Proposition C.10.

**Fact C.6.** *Let $x_o \in \mathcal{X}$, $\alpha, \beta \in \mathcal{M}_1^+(\mathcal{X})$ and $(u,v) \in \mathcal{C}(\mathcal{X})^2$ be the pair of corresponding Sinkhorn potentials with $u(x_o) = 0$. The function $\mathrm{OT}_\varepsilon$ is directionally differentiable and the directional derivative of $\mathrm{OT}_\varepsilon$ in $(\alpha, \beta)$ along a feasible direction $(\mu, \nu) \in \mathcal{F}_{\mathcal{M}_1^+(\mathcal{X})^2}((\alpha,\beta))$ (see Definition A.2) is*

$$\mathrm{OT}'_\varepsilon(\alpha, \beta; \mu, \nu) = \int u(x)\,d\mu(x) + \int v(y)\,d\nu(y) = \langle (u,v), (\mu,\nu) \rangle. \tag{C.17}$$

*Let $\nabla\mathrm{OT}_\varepsilon \colon \mathcal{M}_1^+(\mathcal{X})^2 \to \mathcal{C}(\mathcal{X})^2$ be the operator that maps every pair of probability distributions $(\alpha, \beta) \in \mathcal{M}_1^+(\mathcal{X})^2$ to the corresponding pair of Sinkhorn potentials $(u, v) \in \mathcal{C}(\mathcal{X})^2$ with $u(x_o) = 0$. Then (C.17) can be written as*

$$\mathrm{OT}'_\varepsilon(\alpha, \beta; \mu, \nu) = \langle \nabla\mathrm{OT}_\varepsilon(\alpha, \beta), (\mu, \nu) \rangle. \tag{C.18}$$

**Remark C.7.** In Fact C.6, the requirement $u(x_o) = 0$ is only a convention to remove ambiguities. Indeed, for every $t \in \mathbb{R}$, replacing the Sinkhorn potential $(u + t, u - t)$ in Definition A.1 does not affect (C.17).

**Fact C.8.** *Let $\beta \in \mathcal{M}_+^1(\mathcal{X})$ and let $\nabla_1 \mathrm{OT}_\varepsilon$ be the first component of the gradient operator defined in Fact C.6. Then the Sinkhorn divergence function $S_\varepsilon(\cdot, \beta) \colon \mathcal{M}_+^1(\mathcal{X}) \to \mathbb{R}$ in (7) is directionally differentiable and, for every $\alpha \in \mathcal{M}_+^1(\mathcal{X})$ and every $\mu \in \mathcal{F}_{\mathcal{M}_+^1(\mathcal{X})}(\alpha)$,*

$$[S_\varepsilon(\cdot, \beta)]'(\alpha; \mu) = \langle \nabla_1 \mathrm{OT}_\varepsilon(\alpha, \beta) - \nabla_1 \mathrm{OT}_\varepsilon(\alpha, \alpha), \mu \rangle.$$

*So, one can define $\nabla S_\varepsilon(\cdot, \beta) \colon \mathcal{M}_1^+(\mathcal{X}) \to \mathcal{C}(\mathcal{X})$ such that, for every $\alpha \in \mathcal{M}_+^1(\mathcal{X})$, $\nabla[S_\varepsilon(\cdot, \beta)](\alpha) = \nabla_1 \mathrm{OT}_\varepsilon(\alpha, \beta) - \nabla_1 \mathrm{OT}_\varepsilon(\alpha, \alpha)$ and we have*

$$[S_\varepsilon(\cdot, \beta)]'(\alpha; \mu) = \langle \nabla S_\varepsilon(\cdot, \beta), \mu \rangle. \tag{C.19}$$

*Finally, if $\mathsf{k}$ in (B.1) is a positive definite kernel, then the Sinkhorn divergence $S_\varepsilon(\cdot, \beta)$ is convex.*

We are now ready to prove Theorem 4 in the paper. We recall also the statement for reader's convenience.

**Theorem 4.** *The gradient $\nabla \mathrm{OT}_\varepsilon$ defined in Proposition 1 is Lipschitz continuous. In particular, the first component $\nabla_1 \mathrm{OT}_\varepsilon$ is $2\varepsilon e^{3\mathsf{D}/\varepsilon}$-Lipschitz continuous, i.e., for every $\alpha, \alpha', \beta, \beta' \in \mathcal{M}_1^+(\mathcal{X})$,*

$$\|u - u'\|_\infty = \|\nabla_1 \mathrm{OT}_\varepsilon(\alpha, \beta) - \nabla_1 \mathrm{OT}_\varepsilon(\alpha', \beta')\|_\infty \leq 2\varepsilon e^{3\mathsf{D}/\varepsilon} \left( \|\alpha - \alpha'\|_{TV} + \|\beta - \beta'\|_{TV} \right), \tag{11}$$

*where $\mathsf{D} = \sup_{x,y \in \mathcal{X}} \mathsf{c}(x,y)$, $u = \mathsf{T}_{\beta\alpha}(u)$, $u' = \mathsf{T}_{\beta'\alpha'}(u')$, and $u(x_o) = u'(x_o) = 0$. Moreover, it follows from (8) that $\nabla S_\varepsilon(\cdot, \beta)$ is $6\varepsilon e^{3\mathsf{D}/\varepsilon}$-Lipschitz continuous. The same holds for $\nabla \mathsf{B}_\varepsilon$.*

*Proof.* The first part is just a consequence of Theorem C.4 and Fact C.6. The second part, follows from the first part and Fact C.8. $\qquad\square$

**Remark C.9.** It follows from the optimality conditions (B.25) that, for every $x \in \mathrm{supp}(\alpha)$ and $y \in \mathrm{supp}(\beta)$,

$$1 = \int_\mathcal{X} e^{\frac{u(x)+v(y)-\mathsf{c}(x,y)}{\varepsilon}} d\beta(y) \quad \text{and} \quad 1 = \int_\mathcal{X} e^{\frac{u(x)+v(y)-\mathsf{c}(x,y)}{\varepsilon}} d\alpha(x),$$

hence,

$$\int_\mathcal{X} e^{\frac{u \oplus v - \mathsf{c}}{\varepsilon}} d\alpha \otimes \beta = 1. \tag{C.20}$$

Then, recalling the definition of $\mathrm{OT}_\varepsilon$ in (2) and that of its gradient, given above, we have

$$\mathrm{OT}_\varepsilon(\alpha, \beta) = \langle \nabla \mathrm{OT}_\varepsilon(\alpha, \beta), (\alpha, \beta) \rangle - \varepsilon. \tag{C.21}$$

Since, $\nabla \mathrm{OT}_\varepsilon$ is bounded and Lipschitz continuous, it follows that $\mathrm{OT}_\varepsilon$ is Lipschitz continuous with respect to the total variation.

We end the section by providing an independent proof of Fact C.6, which is based on Proposition A.8 and Corollary C.5.

**Proposition C.10.** *The function $\mathrm{OT}_\varepsilon \colon \mathcal{M}_+^1(\mathcal{X})^2 \to \mathbb{R}$, defined in (2), is continuous with respect to the total variation, directionally differentiable, and, for every $(\alpha, \beta) \in \mathcal{M}_+^1(\mathcal{X})^2$ and every feasible direction $(\mu, \nu) \in \mathcal{F}_{\mathcal{M}_+^1(\mathcal{X})^2}(\alpha, \beta)$, we have*

$$\mathrm{OT}_\varepsilon'(\alpha, \beta; \mu, \nu) = \langle (u, v), (\mu, \nu) \rangle, \tag{C.22}$$

*where $(u, v) \in \mathcal{C}(\mathcal{X})^2$ is any solution of problem (2).*

*Proof.* Let $g \colon \mathcal{C}(\mathcal{X})^2 \times \mathcal{M}(\mathcal{X})^2 \to \mathbb{R}$ be such that,

$$g((u, v), (\alpha, \beta)) = \langle u, \alpha \rangle + \langle v, \beta \rangle - \varepsilon \langle \exp((u \oplus v - \mathsf{c})/\varepsilon), \alpha \otimes \beta \rangle. \tag{C.23}$$

Then, for every $(\alpha, \beta) \in \mathcal{M}_+^1(\mathcal{X})^2$,

$$\mathrm{OT}_\varepsilon(\alpha, \beta) = \max_{(u,v) \in \mathcal{C}(\mathcal{X})^2} g((u, v), (\alpha, \beta)). \tag{C.24}$$

Thus, $\text{OT}_\varepsilon$ is of the type considered in Proposition A.8. Let $(u,v) \in \mathcal{C}(\mathcal{X})$. Then the function $g((u,v),\cdot)$ admits directional derivatives and, for every $(\alpha,\beta),(\mu,\nu) \in \mathcal{M}(\mathcal{X})^2$, we have

$$[g((u,v),\cdot)]'((\alpha,\beta);(\mu,\nu))$$
$$= \left\langle u - \varepsilon e^{\frac{u}{\varepsilon}} \int_{\mathcal{X}} e^{\frac{v-c(\cdot,y)}{\varepsilon}} d\beta(y), \mu \right\rangle + \left\langle v - \varepsilon e^{\frac{v}{\varepsilon}} \int_{\mathcal{X}} e^{\frac{u-c(x,\cdot)}{\varepsilon}} d\alpha(x), \nu \right\rangle. \quad \text{(C.25)}$$

Indeed, for every $t > 0$,

$$\frac{1}{t}\big[g((u,v),(\alpha,\beta)+t(\mu,\nu)) - g((u,v),(\alpha,\beta))\big]$$
$$= \frac{1}{t}\big[\langle u, \alpha + t\mu\rangle + \langle v, \beta + t\nu\rangle - \varepsilon\langle \exp((u \oplus v - \mathsf{c})/\varepsilon), (\alpha + t\mu) \otimes (\beta + t\nu)\rangle$$
$$- \langle u, \alpha\rangle - \langle v, \beta\rangle + \varepsilon\langle \exp((u \oplus v - \mathsf{c})/\varepsilon), \alpha \otimes \beta\rangle\big]$$
$$= \langle u, \mu\rangle + \langle v, \nu\rangle - \varepsilon\langle \exp((u \oplus v - \mathsf{c})/\varepsilon), \alpha \otimes \nu\rangle - \varepsilon\langle \exp((u \oplus v - \mathsf{c})/\varepsilon), \mu \otimes \beta\rangle$$
$$- t\varepsilon\langle \exp((u \oplus v - \mathsf{c})/\varepsilon), \mu \otimes \nu\rangle,$$

hence

$$[g((u,v),\cdot)]'((\alpha,\beta);(\mu,\nu))$$
$$= \langle u, \mu\rangle + \langle v, \nu\rangle - \varepsilon\langle \exp((u \oplus v - \mathsf{c})/\varepsilon), \alpha \otimes \nu\rangle - \varepsilon\langle \exp((u \oplus v - \mathsf{c})/\varepsilon), \mu \otimes \beta\rangle$$

and (C.25) follows. Thus, the function $g$ is Gâteaux differentiable with respect to the second variable, with derivative

$$D_2 g((u,v),(\alpha,\beta)) = \left(u - \varepsilon e^{\frac{u}{\varepsilon}} \int_{\mathcal{X}} e^{\frac{v-c(\cdot,y)}{\varepsilon}} d\beta(y), v - \varepsilon e^{\frac{v}{\varepsilon}} \int_{\mathcal{X}} e^{\frac{u-c(x,\cdot)}{\varepsilon}} d\alpha(x)\right)$$
$$= (u,v) - \varepsilon(e^{\frac{u}{\varepsilon}}\mathsf{L}_\beta e^{\frac{v}{\varepsilon}}, e^{\frac{v}{\varepsilon}}\mathsf{L}_\alpha e^{\frac{u}{\varepsilon}}) \in \mathcal{C}(\mathcal{X})^2,$$

which is jointly continuous, since the maps $(u,\alpha) \mapsto \mathsf{L}_\alpha e^{u/\varepsilon}$ and $(v,\beta) \mapsto \mathsf{L}_\beta e^{v/\varepsilon}$ are continuous. Moreover, it follows from Corollary C.5 that there exists a continuous selection of Sinkhorn potentials. Therefore, it follows from Proposition A.8 that $\text{OT}_\varepsilon$ is directionally differentiable and

$$\text{OT}_\varepsilon'((\alpha,\beta);(\mu,\nu)) = \max_{(u,v) \text{ solution of (C.24)}} \langle D_2 g((u,v),(\alpha,\beta)),(\mu,\nu)\rangle. \quad \text{(C.26)}$$

However, if $(u,v)$ is a solution of (C.24), it follows from the optimality conditions (B.25) that

$$e^{\frac{u}{\varepsilon}} \int_{\mathcal{X}} e^{\frac{v-c(\cdot,y)}{\varepsilon}} d\beta(y) = 1 \quad \text{and} \quad e^{\frac{v}{\varepsilon}} \int_{\mathcal{X}} e^{\frac{u-c(x,\cdot)}{\varepsilon}} d\alpha(x) = 1, \quad \text{(C.27)}$$

hence

$$\langle D_2 g((u,v),(\alpha,\beta)),(\mu,\nu)\rangle = \langle(u-\varepsilon, v-\varepsilon),(\mu,\nu)\rangle = \langle(u,v),(\mu,\nu)\rangle, \quad \text{(C.28)}$$

where we used the fact that, since $(\mu,\nu) = t(\mu_1 - \mu_2, \nu_1 - \nu_2)$ for some $t > 0$ and $\mu_1, \mu_2, \nu_1, \nu_2 \in \mathcal{M}_+^1(\mathcal{X})$, we have $\langle 1, \mu\rangle = t\langle 1, \mu_1 - \mu_2\rangle = 0$ and $\langle 1, \nu\rangle = t\langle 1, \nu_1 - \nu_2\rangle = 0$. $\qquad\square$

## D  The Frank-Wolfe algorithm for Sinkhorn barycenters

In this section we finally analyze the Frank-Wolfe algorithm for the Sinkhorn barycenters and give convergence results. The following result is a direct consequence of Theorem B.10 and Fact C.6.

**Theorem D.1.** *Let $(\tilde{u}^{(\ell)})_{\ell \in \mathbb{N}}$ be generated through Algorithm B.1 as in Theorem B.10. Then,*

$$(\forall \ell \in \mathbb{N}) \quad \|\tilde{u}^{(\ell)} - \nabla_1 \text{OT}_\varepsilon(\alpha,\beta)\|_\infty \leq \lambda^{2\ell}\left(\frac{\mathsf{D} + \max_{\mathcal{X}} u^{(0)} - \min_{\mathcal{X}} u^{(0)}}{\varepsilon}\right), \quad \text{(D.1)}$$

*where $u^{(\ell)} = \varepsilon \log f^{(\ell)}$ and $\tilde{u}^{(\ell)} = u^{(\ell)} - u^{(\ell)}(x_o)$.*

Therefore, in view of Fact C.8, Theorem D.1, and Proposition A.7, we can address the problem of the Sinkhorn barycenter (9) via the Frank-Wolfe Algorithm A.1. Note that, according to Proposition A.7(ii), since the diameter of $\mathcal{M}_1^+(\mathcal{X})$ with respect to $\|\cdot\|_{TV}$ is 2, we have that the curvature of $\mathsf{B}_\varepsilon$ is upper bounded by

$$C_{\mathsf{B}_\varepsilon} \leq 24\varepsilon e^{3\mathsf{D}/\varepsilon}. \tag{D.2}$$

Let $k \in \mathbb{N}$ and $\alpha_k$ be the current iteration. For every $j \in \{1,\dots,m\}$, we can compute $\nabla_1 \mathsf{OT}_\varepsilon(\alpha_k,\beta_j)$ and $\nabla_1 \mathsf{OT}_\varepsilon(\alpha_k,\alpha_k)$ by the Sinkhorn-Knopp algorithm. Thus, by (D.1), we find $\ell \in \mathbb{N}$ large enough so that $\|\tilde{u}_j^{(\ell)} - \nabla_1 \mathsf{OT}_\varepsilon(\alpha_k,\beta_q)\|_\infty \leq \Delta_{1,k}/8$ and $\|\tilde{p}^{(\ell)} - \nabla_1 \mathsf{OT}_\varepsilon(\alpha_k,\alpha_k)\|_\infty \leq \Delta_{1,k}/8$ and we set

$$\tilde{u}^{(\ell)} := \sum_{j=1}^m \omega_j \tilde{u}_j^{(\ell)} - \tilde{p}^{(\ell)}. \tag{D.3}$$

Then,

$$\|\tilde{u}^{(\ell)} - \nabla\mathsf{B}_\varepsilon(\alpha_k)\|_\infty \leq \frac{\Delta_{1,k}}{4}. \tag{D.4}$$

Now, Frank-Wolf Algorithm A.1 (in the version considered in Proposition A.7(i)) requires finding

$$\eta_{k+1} \in \operatorname*{argmin}_{\eta \in \mathcal{M}_1^+(\mathcal{X})} \langle \tilde{u}^{(\ell)}, \eta - \alpha_k \rangle \tag{D.5}$$

and make the update

$$\alpha_{k+1} = (1-\gamma_k)\alpha_k + \gamma_k \eta_{k+1}. \tag{D.6}$$

Since the solution of (D.5) is a Dirac measure (see Section 4 in the paper), the algorithm reduces to

$$\begin{cases} \text{find } x_{k+1} \in \mathcal{X} \text{ such that } \tilde{u}^{(\ell)}(x_{k+1}) \leq \min_{x \in \mathcal{X}} \tilde{u}^{(\ell)}(x) + \dfrac{\Delta_{2,k}}{2} \\ \alpha_{k+1} = (1-\gamma_k)\alpha_k + \gamma_k \delta_{x_{k+1}}. \end{cases} \tag{D.7}$$

So, if we initialize the algorithm with $\alpha_0 = \delta_{x_0}$, then any $\alpha_k$ will be a discrete probability measure with support contained in $\{x_0,\dots,x_k\}$. This implies that if all the $\beta_j$'s are probability measures with finite support, the computation of $\nabla_1 \mathsf{OT}_\varepsilon(\alpha_k,\beta_j)$ by the Sinkhorn algorithm can be reduced to a fully discrete algorithm, as showed in Proposition B.11. More precisely, assume that

$$(\forall\, j=1,\dots,m) \quad \beta_j = \sum_{i_2=0}^n b_{j,i_2} \delta_{y_{j,i_2}}. \tag{D.8}$$

and that at iteration $k$ we have

$$\alpha_k = \sum_{i_1=0}^k a_{k,i_1} \delta_{x_{i_1}}. \tag{D.9}$$

Set

$$\mathsf{a}_k = \begin{bmatrix} a_{k,0} \\ \vdots \\ a_{k,k} \end{bmatrix} \in \mathbb{R}^{k+1}, \quad \mathsf{M}_{0,k} = \begin{bmatrix} a_{k,0}\mathsf{k}(x_0,x_0)a_{k,0} & \dots & a_{k,0}\mathsf{k}(x_0,x_k)a_{k,k} \\ \vdots & \ddots & \vdots \\ a_{k,k}\mathsf{k}(x_k,x_0)a_{k,0} & \dots & a_{k,k}\mathsf{k}(x_k,x_k)a_{k,k} \end{bmatrix} \in \mathbb{R}^{(k+1)\times(k+1)} \tag{D.10}$$

and, for every $j = 1\dots,m$,

$$\mathsf{b}_j = \begin{bmatrix} b_{j,0} \\ \vdots \\ b_{j,n} \end{bmatrix} \in \mathbb{R}^{n+1}, \quad \mathsf{M}_{j,k} = \begin{bmatrix} a_{k,0}\mathsf{k}(x_0,y_{j,0})b_{j,0} & \dots & a_{k,0}\mathsf{k}(x_0,y_{j,n})b_{j,n} \\ \vdots & \ddots & \vdots \\ a_{k,k}\mathsf{k}(x_k,y_{j,0})b_{j,0} & \dots & a_{k,n}\mathsf{k}(x_k,y_{j,n})b_{j,n} \end{bmatrix} \in \mathbb{R}^{(k+1)\times(n+1)}. \tag{D.11}$$

Then, run Algorithm B.2, with input $\mathsf{a}_k$, $\mathsf{a}_k$, and $\mathsf{M}_{0,k}$ to get $(\mathsf{e}^{(\ell)},\mathsf{h}^{(\ell)})$, and, for every $j = 1,\dots,m$, with input $\mathsf{a}_k$, $\mathsf{b}_j$, and $\mathsf{M}_{j,k}$ to get $(\mathsf{f}_j^{(\ell)},\mathsf{g}_j^{(\ell)})$. So, we have,

$$(\forall\,\ell \in \mathbb{N}) \quad \begin{cases} \mathsf{h}^{(\ell+1)} = \dfrac{\mathsf{a}_k}{\mathsf{M}_{0,k}^\top \mathsf{e}^{(\ell)}}, \quad \mathsf{e}^{(\ell+1)} = \dfrac{\mathsf{a}_k}{\mathsf{M}_{0,k}\mathsf{h}^{(\ell+1)}} \\ (\forall\, j=1,\dots,m)\ \mathsf{g}_j^{(\ell+1)} = \dfrac{\mathsf{b}_j}{\mathsf{M}_{j,k}^\top \mathsf{f}_j^{(\ell)}}, \quad \mathsf{f}_j^{(\ell+1)} = \dfrac{\mathsf{a}_k}{\mathsf{M}_{j,k}\mathsf{g}_j^{(\ell+1)}}. \end{cases} \tag{D.12}$$

Then, according to Proposition B.11, for every $\ell \in \mathbb{N}$, we have

$$
(\forall\, x \in \mathcal{X}) \quad
\begin{cases}
e^{(\ell)}(x)^{-1} = \displaystyle\sum_{i_2=0}^{k} \mathsf{k}(x, x_{i_2})\mathsf{h}_{i_2}^{(\ell-1)} a_{k,i_2}, \\[3mm]
p^{(\ell)}(x) = \varepsilon \log e^{(\ell)}(x) = -\varepsilon \log \displaystyle\sum_{i_2=0}^{k} \mathsf{k}(x, x_{i_2})\mathsf{h}_{i_2}^{(\ell-1)} a_{k,i_2} \\[3mm]
\tilde{p}^{(\ell)}(x) = p^{(\ell)}(x) - p^{(\ell)}(x_o).
\end{cases}
\tag{D.13}
$$

and, for every $j = 1, \ldots, m$,

$$
(\forall\, x \in \mathcal{X}) \quad
\begin{cases}
f_j^{(\ell)}(x)^{-1} = \displaystyle\sum_{i_2=0}^{n} \mathsf{k}(x, y_{i_2})\mathsf{g}_{j,i_2}^{(\ell-1)} b_{j,i_2}, \\[3mm]
u_j^{(\ell)}(x) = \varepsilon \log f_j^{(\ell)}(x) = -\varepsilon \log \displaystyle\sum_{i_2=0}^{n} \mathsf{k}(x, y_{i_2})\mathsf{g}_{j,i_2}^{(\ell-1)} b_{j,i_2} \\[3mm]
\tilde{u}_j^{(\ell)}(x) = u_j^{(\ell)}(x) - u_j^{(\ell)}(x_o).
\end{cases}
\tag{D.14}
$$

Since the $\tilde{u}_j^{(\ell)}$'s and $u_j^{(\ell)}$'s, and $\tilde{p}^{(\ell)}$ and $p^{(\ell)}$, differ for a constant only, the final algorithm can be written as in Algorithm D.1. We stress that this algorithm is even more general than Algorithm 2 since, in the computation of the Sinkhorn potentials and in their minimization, errors have been taken into account.

---

**Algorithm D.1** Frank-Wolfe algorithm for Sinkhorn barycenter

---

Let $\alpha_0 = \delta_{x_0}$ for some $x_0 \in \mathcal{X}$. Let $(\Delta_{1,k})_{k\in\mathbb{N}}, (\Delta_{2,k})_{k\in\mathbb{N}} \in \mathbb{R}_+^{\mathbb{N}}$ be such that $(2\Delta_{1,k} + \Delta_{2,k})/\gamma_k$ is nondecreasing. Define

> for $k = 0, 1, \ldots$
>> run Algorithm B.2 with input $\mathsf{a}_k, \mathsf{a}_k, \mathsf{M}_{0,k}$ till $\lambda^{2\ell}\mathsf{D}/\varepsilon \leq \frac{\Delta_{1,k}}{8} \;\rightarrow\; \mathsf{h} \in \mathbb{R}^{k+1}$
>> compute $p$ via (D.13) with $\mathsf{h}$
>> for $j = 1, \ldots m$
>>> run Algorithm B.2 with input $\mathsf{a}_k, \mathsf{b}_j, \mathsf{M}_{j,k}$ till $\lambda^{2\ell}\mathsf{D}/\varepsilon \leq \frac{\Delta_{1,k}}{8} \;\rightarrow\; \mathsf{g}_j \in \mathbb{R}^{n+1}$
>>> compute $u_j$ via (D.14) with $\mathsf{g}_j$
>> set $u = \sum_{j=1}^{m} \omega_j u_j - p$
>> find $x_{k+1} \in \mathcal{X}$ such that $u(x_{k+1}) \leq \min_{x\in\mathcal{X}} u(x) + \dfrac{\Delta_{2,k}}{2}$
>> $\alpha_{k+1} = (1 - \gamma_k)\alpha_k + \gamma_k \delta_{x_{k+1}}.$

---

We now give a final converge theorem, of which Theorem 5 in the paper is a special case.

**Theorem D.2.** *Suppose that* $\beta_1, \ldots, \beta_m \in \mathcal{M}_+^1(\mathcal{X})$ *are probability measures with finite support, each of cardinality* $n \in \mathbb{N}$. *Let* $(\alpha_k)_{k\in\mathbb{N}}$ *be generated by Algorithm D.1. Then, for every* $k \in \mathbb{N}$,

$$
\mathsf{B}_\varepsilon(\alpha_k) - \min_{\alpha\in\mathcal{M}_+^1(\mathcal{X})} \mathsf{B}_\varepsilon(\alpha) \leq \gamma_k 24\varepsilon e^{3\mathsf{D}/\varepsilon} + 2\Delta_{1,k} + \Delta_{2,k}
\tag{D.15}
$$

*Proof.* It follows from Theorem A.5, (D.2), and Proposition A.7, recalling that $\mathrm{diam}(\mathcal{M}_+^1(\mathcal{X})) = 2$. $\qquad\square$

# E  Sample complexity of Sinkhorn potential

In the following we will denote by $\mathcal{C}^s(\mathcal{X})$ the space of $s$-differentiable functions with continuous derivatives and by $W^{s,p}(\mathcal{X})$ the Sobolev space of functions $f\colon \mathcal{X} \to \mathbb{R}$ with $p$-summable weak derivatives up to order $s$ [1]. We denote by $\|\cdot\|_{s,p}$ the corresponding norm.

The following result shows that under suitable smoothness assumptions on the cost function c, the Sinkhorn potentials are uniformly bounded as functions in a suitable Sobolev space of corresponding smoothness. This fact will play a key role in approximating the Sinkhorn potentials of general distributions in practice.

**Theorem E.1** (Proposition 2 in [23]). *Let $\mathcal{X}$ be a closed bounded domain with Lipschitz boundary in $\mathbb{R}^d$ ([1, Definition 4.9]) and let $\mathsf{c} \in \mathcal{C}^{s+1}(\mathcal{X} \times \mathcal{X})$. Then for every $(\alpha, \beta) \in \mathcal{M}_1^+(\mathcal{X})^2$, the associated Sinkhorn potentials $(u, v) \in \mathcal{C}(\mathcal{X})^2$ are functions in $W^{s,\infty}(\mathcal{X})$. Moreover, let $x_o \in \mathcal{X}$. Then there exists a constant $\mathsf{r} > 0$, depending only on $\varepsilon, s$ and $\mathcal{X}$, such that for every $(\alpha, \beta) \in \mathcal{M}_1^+(\mathcal{X})^2$ the associated Sinkhorn potentials $(u, v) \in \mathcal{C}(\mathcal{X})^2$ with $u(x_o) = 0$ satisfies $\|u\|_{s,\infty}, \|v\|_{s,\infty} \leq \mathsf{r}$.*

In the original statement of [23, Proposition 2] the above result is formulated for $\mathsf{c} \in \mathcal{C}^{\infty}(\mathcal{X})$ for simplicity. However, as clarified by the authors, it holds also for the more general case $\mathsf{c} \in \mathcal{C}^{s+1}(\mathcal{X})$.

**Lemma E.2.** *Let $\mathcal{X} \subset \mathbb{R}^d$ be a closed bounded domain with Lipschitz boundary and let $u, u' \in W^{s,\infty}(\mathcal{X})$. Then the following holds*

    (i) $\|uu'\|_{s,\infty} \leq \mathsf{m}_1 \|u\|_{s,\infty} \|u'\|_{s,\infty}$,

    (ii) $\|e^u\|_{s,\infty} \leq \|e^u\|_{\infty} (1 + \mathsf{m}_2 \|u\|_{s,\infty})$,

*where $\mathsf{m}_1 = \mathsf{m}_1(s,d)$ and $\mathsf{m}_2 = \mathsf{m}_2(s,d) > 0$ depend only on the dimension $d$ and the order of differentiability $s$ but not on $u$ and $u'$.*

*Proof.* (i) follows directly from Leibniz formula. To see (ii), let $\mathbf{i} = (i_1, \ldots, i_d) \in \mathbb{N}^d$ be a multi-index with $|\mathbf{i}| = \sum_{\ell=1}^{d} i_\ell \leq s$ and note that by chain rule the derivatives of $e^u$

$$D^{\mathbf{i}} e^u = e^u P_{\mathbf{i}}\Big((D^{\mathbf{j}}u)_{\mathbf{j}\leq\mathbf{i}}\Big),$$

where $P_{\mathbf{i}}$ is a polynomial of degree $|\mathbf{i}|$ and $\mathbf{j} \leq \mathbf{i}$ is the ordering associated to the cone of non-negative vectors in $\mathbb{R}^d$. Note that $P_0 = 1$, while for $|\mathbf{i}| > 0$, the associated polyomial $P_{\mathbf{i}}$ has a root in zero (i.e. it does not have constant term). Hence

$$\|e^u\|_{s,\infty} \leq \|e^u\|_{\infty} \left(1 + |P|\Big((\|D^{\mathbf{i}}u\|_{\infty})_{|\mathbf{i}|\leq s}\Big)\right),$$

where we have denoted by $P = \sum_{0<|\mathbf{i}|\leq s} P_{\mathbf{i}}$ and by $|P|$ the polynomial with coefficients corresponding to the absolute value of the coefficients of $P$. Therefore, since $\|D^{\mathbf{i}}u\|_{\infty} \leq \|u\|_{s,\infty}$ for any $|\mathbf{i}| \leq s$, by taking

$$\mathsf{m}_2 = |P|\Big((1)_{|\mathbf{i}|\leq s}\Big),$$

namely the sum of all the coefficients of $|P|$, we obtain the desired result. Indeed note that the coefficients of $P$ do not depend on $u$ but only on the smoothness $s$ and dimension $d$. $\qquad\square$

**Lemma E.3.** *Let $\mathcal{X} \subset \mathbb{R}^d$ be a closed bounded domain with Lipschitz boundary and let $x_o \in \mathcal{X}$. Let $\mathsf{c} \in \mathcal{C}^{s+1}(\mathcal{X} \times \mathcal{X})$, for some $s \in \mathbb{N}$. Then for any $\alpha, \beta \in \mathcal{M}_1^+(\mathcal{X})$ and corresponding pair of Sinkhorn potentials $(u, v) \in \mathcal{C}(\mathcal{X})^2$ with $u(x_o) = 0$, the functions $\mathsf{k}(x, \cdot)e^{u/\varepsilon}$ and $\mathsf{k}(x, \cdot)e^{v/\varepsilon}$ belong to $W^{s,2}(\mathcal{X})$ for every $x \in \mathcal{X}$. Moreover, they admit an extension to $\mathcal{H} = W^{s,2}(\mathbb{R}^d)$ and there exists a constant $\bar{\mathsf{r}}$ independent on $\alpha$ and $\beta$, such that for every $x \in \mathcal{X}$*

$$\left\|\mathsf{k}(x, \cdot)e^{u/\varepsilon}\right\|_{\mathcal{H}}, \left\|\mathsf{k}(x, \cdot)e^{v/\varepsilon}\right\|_{\mathcal{H}} \leq \bar{\mathsf{r}} \tag{E.1}$$

*(with some abuse of notation, we have identified $\mathsf{k}(x, \cdot)e^{u/\varepsilon}$ and $\mathsf{k}(x, \cdot)e^{v/\varepsilon}$ with their extensions to $\mathbb{R}^d$).*

*Proof.* In the following we denote by $\|\cdot\|_{s,2} = \|\cdot\|_{s,2,\mathcal{X}}$ the norm of $W^{s,2}(\mathcal{X})$ and by $\|\cdot\|_{\mathcal{H}} = \|\cdot\|_{s,2,\mathbb{R}^d}$ the norm of $\mathcal{H} = W^{s,2}(\mathbb{R})$. Let $x \in \mathcal{X}$. Then, since $u - \mathsf{c}(x, \cdot) \in W^{s,\infty}(\mathcal{X})$ and

$\|u\|_{s,\infty} \leq \mathsf{r}$, it follows from Lemma E.2 that

$$
\begin{aligned}
\left\|\mathsf{k}(x,\cdot)e^{u/\varepsilon}\right\|_{s,\infty} &= \left\|e^{(u-\mathsf{c}(x,\dot{)})/\varepsilon}\right\|_{s,\infty} \\
&\leq \left\|e^{(u-\mathsf{c}(x,\dot{)})/\varepsilon}\right\|_{\infty}(1+\mathsf{m}_2\left\|u-\mathsf{c}(x,\cdot)\right\|_{s,\infty}) \\
&= \left\|\mathsf{k}(x,\cdot)e^{u/\varepsilon}\right\|_{\infty}(1+\mathsf{m}_2\left\|u-\mathsf{c}(x,\cdot)\right\|_{s,\infty}) \\
&\leq \left\|e^{u/\varepsilon}\right\|_{\infty}(1+\mathsf{m}_2(\mathsf{r}+\left\|\mathsf{c}\right\|_{s,\infty})) \\
&\leq e^{\mathsf{D}/\varepsilon}(1+\mathsf{m}_2(\mathsf{r}+\left\|\mathsf{c}\right\|_{s,\infty})),
\end{aligned}
$$

where we used the fact that $D^{\mathbf{i}}[\mathsf{c}(x,\cdot)] = (D^{\mathbf{i}}\mathsf{c})(x,\cdot)$. This implies

$$
\left\|\mathsf{k}(x,\cdot)e^{u/\varepsilon}\right\|_{s,2} \leq |\mathcal{X}|^{1/2}e^{\mathsf{D}/\varepsilon}(1+\mathsf{m}_2(\mathsf{r}+\left\|\mathsf{c}\right\|_{s,\infty}))
$$

where $|\mathcal{X}|$ is the Lebesgue measure of $\mathcal{X}$. Now, we can proceed analogously to [23, Proposition 2], and use Stein's Extension Theorem [1, Theorem 5.24],[51, Chapter 6], to guarantee the existence of a *total extension operator* [1, Definition 5.17]. In particular, there exists a constant $\mathsf{m}_3 = \mathsf{m}_3(s,2,\mathcal{X})$ such that for any $\varphi \in W^{s,2}(\mathcal{X})$ there exists $\tilde{\varphi} \in W^{s,2}(\mathbb{R}^d)$ such that

$$
\|\tilde{\varphi}\|_{\mathcal{H}} = \|\tilde{\varphi}\|_{s,2,\mathbb{R}^d} \leq \mathsf{m}_3\|\varphi\|_{s,2,\mathcal{X}} = \mathsf{m}_3\|\varphi\|_{s,2}. \tag{E.2}
$$

Therefore, we conclude

$$
\left\|\mathsf{k}(x,\cdot)e^{u/\varepsilon}\right\|_{\mathcal{H}} \leq \mathsf{m}_3|\mathcal{X}|^{1/2}e^{\mathsf{D}/\varepsilon}(1+\mathsf{m}_2(\mathsf{r}+\left\|\mathsf{c}\right\|_{s,\infty})) =: \bar{\mathsf{r}}. \tag{E.3}
$$

The same argument applies to $\mathsf{k}(x,\cdot)e^{v/\varepsilon}$ with the only exception that now, in virtue of Corollary B.9, we have $\|e^{v/\varepsilon}\|_{\infty} \leq e^{2\mathsf{D}/\varepsilon}$. Note that $\bar{\mathsf{r}}$ is a constant depending only on $\mathcal{X}$, $\mathsf{c}$, $s$ and $d$ but it is independent on the probability distributions $\alpha$ and $\beta$. $\qquad\square$

**Sobolev spaces and reproducing kernel Hilbert spaces.** Recall that for $s > d/2$ the space $\mathcal{H} = W^{s,2}(\mathbb{R}^d)$, is a reproducing kernel Hilbert space (RKHS) [53, Chapter 10]. In this setting we denote by $\mathsf{h} : \mathcal{X} \times \mathcal{X} \to \mathbb{R}$ the associated reproducing kernel, which is continuous and bounded and satisfies the reproducing property

$$
(\forall x \in \mathcal{X})(\forall f \in \mathcal{H}) \qquad \langle f, \mathsf{h}(x,\cdot)\rangle_{\mathcal{H}} = f(x). \tag{E.4}
$$

We can also assume that $\mathsf{h}$ is *normalized*, namely, $\|\mathsf{h}(x,\cdot)\|_{\mathcal{H}} = 1$ for all $x \in \mathcal{X}$ [53, Chapter 10].

**Kernel mean embeddings.** For every $\beta \in \mathcal{M}_1^+(\mathcal{X})$, we denote by $\mathsf{h}_\beta \in \mathcal{H}$ the *Kernel Mean Embedding* of $\beta$ in $\mathcal{H}$ [35, 43], that is, the vector

$$
\mathsf{h}_\beta = \int \mathsf{h}(x,\cdot)\,d\beta(x). \tag{E.5}
$$

In other words, the kernel mean embedding of a distribution $\beta$ corresponds to the expectation of $\mathsf{h}(x,\cdot)$ with respect to $\beta$. By the linearity of the inner product and the integral, for every $f \in \mathcal{H}$, the inner product

$$
\langle f, \mathsf{h}_\beta\rangle_{\mathcal{H}} = \int \langle f, \mathsf{h}(x,\cdot)\rangle\,d\beta(x) = \int f(x)\,d\beta(x), \tag{E.6}
$$

corresponds to the expectation of $f(x)$ with respect to $\beta$. The *Maximum Mean Discrepancy (MMD)* [35, 46, 47] between two probability distributions $\beta, \beta' \in \mathcal{M}_1^+(\mathcal{X})$ is defined as

$$
\mathrm{MMD}(\beta, \beta') = \|\mathsf{h}_\beta - \mathsf{h}_{\beta'}\|_{\mathcal{H}}. \tag{E.7}
$$

In the case of the Sobolev space $\mathcal{H} = W^{s,2}(\mathbb{R}^d)$, the MMD metrizes the weak-$*$ topology of $\mathcal{M}_1^+(\mathcal{X})$ [47, 48].

A well-established approach to approximate a distribution $\beta \in \mathcal{M}_1^+(\mathcal{X})$ is to independently sample a set of points $x_1, \ldots, x_n \in \mathcal{X}$ from $\beta$ and consider the empirical distribution $\beta_n = \frac{1}{n}\sum_{i=1}^n \delta_{x_i}$. The following result shows that $\beta_n$ converges to $\beta$ in MMD with high probability. The original version of this result can be found in [46], we report an independent proof for completeness.

**Lemma E.4.** *Let $\beta \in \mathcal{M}_1^+(\mathcal{X})$. Let $x_1, \ldots, x_n \in \mathcal{X}$ be indepedently sampled according to $\beta$ and denote by $\beta_n = \frac{1}{n} \sum_{i=1}^n \delta_{x_i}$. Then, for any $\tau \in (0, 1]$, we have*

$$\mathrm{MMD}(\beta_n, \beta) \leq \frac{4 \log \frac{3}{\tau}}{\sqrt{n}} \tag{E.8}$$

*with probability at least $1 - \tau$.*

*Proof.* The proof follows by applying Pinelis' inequality [39, 42, 55] for random vectors in Hilbert spaces. More precisely, for $i = 1, \ldots, n$, denote by $\zeta_i = \mathsf{h}(x_i, \cdot) \in \mathcal{H}$ and recall that $\|\zeta_i\| = \|\mathsf{h}(x, \cdot)\| = 1$ for all $x \in \mathcal{X}$. We can therefore apply [42, Lemma 2] with constants $\widetilde{M} = 1$ and $\sigma^2 = \sup_i \mathbb{E}\|\zeta_i\|^2 \leq 1$, which guarantees that, for every $\tau \in (0, 1]$

$$\left\| \frac{1}{n} \sum_{i=1}^n \left[ \zeta_i - \mathbb{E}\,\zeta_i \right] \right\|_{\mathcal{H}} \leq \frac{2 \log \frac{2}{\tau}}{n} + \sqrt{\frac{2 \log \frac{2}{\tau}}{n}} \leq \frac{4 \log \frac{3}{\tau}}{\sqrt{n}}, \tag{E.9}$$

holds with probability at least $1 - \tau$. Here, for the second inequality we have used the fact that $\log \frac{2}{\tau} \leq \log \frac{3}{\tau}$ and $\log \frac{3}{\tau} \geq 1$ for every $\tau \in (0, 1]$. The desired result follows by observing that

$$\mathsf{h}_\beta = \int \mathsf{h}(x, \cdot)\, d\beta(x) = \mathbb{E}\,\zeta_i \tag{E.10}$$

for all $i = 1, \ldots, n$, and

$$\mathsf{h}_{\beta_n} = \frac{1}{n} \sum_{i=1}^n \mathsf{h}(x_i, \cdot) = \frac{1}{n} \sum_{i=1}^n \zeta_i. \tag{E.11}$$

Therefore,

$$\mathrm{MMD}(\beta_n, \beta) = \|\mathsf{h}_{\beta_n} - \mathsf{h}_\beta\|_{\mathcal{H}} = \left\| \frac{1}{n} \sum_{i=1}^n \left[ \zeta_i - \mathbb{E}\,\zeta_i \right] \right\|_{\mathcal{H}}, \tag{E.12}$$

which combined with (E.9) leads to the desired result. $\qquad\square$

**Proposition E.5** (Lipschitz continuity of the Sinkhorn Potentials with respect to the MMD)**.** *Let $\mathcal{X} \subset \mathbb{R}^d$ be a compact Lipschitz domain and $\mathsf{c} \in \mathcal{C}^{s+1}(\mathcal{X} \times \mathcal{X})$, with $s > d/2$. Let $\alpha, \beta, \alpha', \beta' \in \mathcal{M}_1^+(\mathcal{X})$. Let $x_o \in \mathcal{X}$ and let $(u, v), (u', v') \in \mathcal{C}(\mathcal{X})^2$ be the two Sinkhorn potentials corresponding to the solution of the regularized OT problem in (B.24) for $(\alpha, \beta)$ and $(\alpha', \beta')$ respectively such that $u(x_o) = u'(x_o) = 0$. Then*

$$\|u - u'\|_\infty \leq 2\varepsilon \bar{r} e^{3\mathrm{D}/\varepsilon} \left( \mathrm{MMD}(\alpha, \alpha') + \mathrm{MMD}(\beta, \beta') \right), \tag{E.13}$$

*with $\bar{r}$ from Lemma E.3. In other words, the operator $\nabla_1 \mathrm{OT}_\varepsilon \colon \mathcal{M}_1^+(\mathcal{X})^2 \to \mathcal{C}(\mathcal{X})$, defined in Fact C.6, is $2\varepsilon \bar{r} e^{3\mathrm{D}/\varepsilon}$-Lipschitz continuous with respect to the* MMD.

*Proof.* Let $f = e^{u/\varepsilon}$ and $g = e^{v/\varepsilon}$. By relying on Lemma E.3 we can now refine the analysis in Theorem C.4. More precisely, we observe that in (C.10) we have

$$\begin{aligned}
[(\mathsf{L}_{\beta'} - \mathsf{L}_\beta)g](x) &= \int \mathsf{k}(x, z)g(z)\, d(\beta - \beta')(z) \\
&= \int \langle \mathsf{k}(x, \cdot)g, \mathsf{h}(z, \cdot) \rangle_{\mathcal{H}}\, d(\beta - \beta')(z) \\
&= \langle \mathsf{k}(x, \cdot)g, \mathsf{h}_\beta - \mathsf{h}_{\beta'} \rangle_{\mathcal{H}} \\
&\leq \|\mathsf{k}(x, \cdot)g\|_{\mathcal{H}}\, \|\mathsf{h}_\beta - \mathsf{h}_{\beta'}\|_{\mathcal{H}} \\
&\leq \bar{r}\, \mathrm{MMD}(\beta, \beta'),
\end{aligned}$$

where in the first equality, with some abuse of notation, we have implicitly considered the extension of $\mathsf{k}(x, \cdot)g$ to $\mathcal{H} = W^{s,2}(\mathbb{R}^d)$ as discussed in Lemma E.3. The rest of the analysis in Theorem C.4 remains invaried, eventually leading to (E.13). $\qquad\square$

It is now clear that Theorem 6 in the paper is just a consequence of Lemma E.4 and Proposition E.5. We give the statement of the theorem for reader's convenience.

**Theorem 6** (Sample Complexity of Sinkhorn Potentials). *Suppose that* $\mathsf{c} \in \mathcal{C}^{s+1}(\mathcal{X} \times \mathcal{X})$ *with* $s > d/2$. *Then, there exists a constant* $\bar{\mathsf{r}} = \bar{\mathsf{r}}(\mathcal{X}, \mathsf{c}, d)$ *such that for any* $\alpha, \beta \in \mathcal{M}_1^+(\mathcal{X})$ *and any empirical measure* $\hat{\beta}$ *of a set of* $n$ *points independently sampled from* $\beta$, *we have, for every* $\tau \in (0, 1]$

$$\|u - u_n\|_\infty = \|\nabla_1 \mathsf{OT}_\varepsilon(\alpha, \beta) - \nabla_1 \mathsf{OT}_\varepsilon(\alpha, \hat{\beta})\|_\infty \leq \frac{8\varepsilon \,\bar{\mathsf{r}} e^{3\mathsf{D}/\varepsilon} \log \frac{3}{\tau}}{\sqrt{n}} \tag{17}$$

*with probability at least* $1 - \tau$, *where* $u = \mathsf{T}_{\beta\alpha}(u), u_n = \mathsf{T}_{\hat{\beta}\alpha}(u_n)$ *and* $u(x_o) = u_n(x_o) = 0$.

We finally provide the proof of Theorem 7 in the paper.

**Theorem 7.** *Suppose that* $\mathsf{c} \in \mathcal{C}^{s+1}(\mathcal{X} \times \mathcal{X})$ *with* $s > d/2$. *Let* $n \in \mathbb{N}$ *and* $\hat{\beta}_1, \ldots, \hat{\beta}_m$ *be empirical distributions with* $n$ *support points, each independently sampled from* $\beta_1, \ldots, \beta_m$. *Let* $\alpha_k$ *be the* $k$-*th iterate of Algorithm 2 applied to* $\hat{\beta}_1, \ldots, \hat{\beta}_m$. *Then for any* $\tau \in (0, 1]$, *the following holds with probability larger than* $1 - \tau$

$$\mathsf{B}_\varepsilon(\alpha_k) - \min_{\alpha \in \mathcal{M}_1^+(\mathcal{X})} \mathsf{B}_\varepsilon(\alpha) \leq \frac{64\bar{\mathsf{r}}\varepsilon e^{3\mathsf{D}/\varepsilon} \log \frac{3m}{\tau}}{\min(k, \sqrt{n})}. \tag{18}$$

*Proof.* Let $\widehat{\mathsf{B}_\varepsilon}(\alpha) = \sum_{j=1}^m \omega_j \mathsf{S}_\varepsilon(\alpha, \hat{\beta}_j)$. We apply Theorem 6 independently for each distribution $\hat{\beta}_j$ and then take the intersection bound between all these separate events. Then, for every $k \in \mathbb{N}$, and with probability larger than $1 - \tau$, we have

$$
\begin{aligned}
\|\nabla \widehat{\mathsf{B}_\varepsilon}(\alpha_k) - \nabla \mathsf{B}_\varepsilon(\alpha_k)\|_\infty &\leq \sum_{j=1}^m \omega_j \|\nabla[\mathsf{S}_\varepsilon(\cdot, \hat{\beta}_j)](\alpha_k) - \mathsf{S}_\varepsilon(\cdot, \beta_j)](\alpha_k)\|_\infty \\
&= \sum_{j=1}^m \omega_j \|\nabla_1 \mathsf{OT}_\varepsilon(\alpha_k, \hat{\beta}_j) - \nabla_1 \mathsf{OT}_\varepsilon(\alpha_k, \beta_j)\|_\infty \\
&\leq \frac{8\varepsilon \,\bar{\mathsf{r}} e^{3\mathsf{D}/\varepsilon} \log \frac{3m}{\tau}}{\sqrt{n}} \\
&= \frac{\Delta_1}{4},
\end{aligned}
$$

where

$$\Delta_1 := \frac{32\varepsilon \,\bar{\mathsf{r}} e^{3\mathsf{D}/\varepsilon} \log \frac{3m}{\tau}}{\sqrt{n}}.$$

Now, let $\gamma_k = 2/(k+2)$. Since Algorithm 2 is applied to $\hat{\beta}_1, \ldots \hat{\beta}_m$, we have

$$\delta_{x_{k+1}} \in \underset{\mathcal{M}_+^1(\mathcal{X})}{\operatorname{argmin}} \langle \nabla \widehat{\mathsf{B}_\varepsilon}(\alpha_k), \cdot \rangle \quad \text{and} \quad \alpha_{k+1} = (1 - \gamma_k)\alpha_k + \gamma_k \delta_{x_{k+1}}.$$

Therefore, it follows from Theorem A.5, Proposition A.7 (with $\Delta_{1,k} = \Delta_1$ and $\Delta_{2,k} = 0$), and Theorem 4 that, with probability larger than $1 - \tau$, we have

$$\mathsf{B}_\varepsilon(\alpha_k) - \min_{\mathcal{M}_+^1(\mathcal{X})} \mathsf{B}_\varepsilon \leq 6\varepsilon\bar{\mathsf{r}} e^{3\mathsf{D}/\varepsilon} \operatorname{diam}(\mathcal{M}_+^1(\mathcal{X}))^2 \gamma_k + \Delta_1 \operatorname{diam}(\mathcal{M}_+^1(\mathcal{X})).$$

The statement follows by noting that $\operatorname{diam}(\mathcal{M}_+^1(\mathcal{X})) = 2$. $\qquad \square$

# F Additional experiments

**Sampling of continuous measures: mixture of Gaussians.** We perform the barycenter of 5 mixtures of two Gaussians $\mu_j$, centered at $(j/2, 1/2)$ and $(j/2, 3/2)$ for $j-0, \dots, 4$ respectively. Samples are provided in Figure 6. We use different relative weights pairs in the mixture of Gaussians, namely $(1/10, 9/10), (1/4, 3/4), (1/2, 1/2)$. At each iteration, a sample of $n = 500$ points is drawn from $\mu_j, j = 0 \dots, 4$. Results are reported in Figure 7.

Fig. 6: Samples of input measures

Fig. 7: Barycenters of Mixture of Gaussians

**Large scale discrete measures: meshes.** We perform the barycenter of two discrete measures with support in $\mathbb{R}^3$. Meshes of the dinosaur are taken from [44] and rescaled by a 0.5 factor. The internal problem in Frank-Wolfe algorithm is solved using L-BFGS-B SciPy optimizer. Formula of the Jacobian is passed to the method. The result is displayed in Figure 8.

**Propagation.** We extend the description on the experiment about propagation in Section 6. Edges $\mathcal{E}$ are selected as follows: we created a matrix $D$ such that $D_{ij}$ contains the distance between station at vertex $i$ and station at vertex $j$, computed using the geographical coordinates of the stations. Each node $v$ in $\mathcal{V}$, is connected to those nodes $u \in \mathcal{V}$ such that $D_{vu} \leq 3$. If the number of nodes $u$ that meet this condition is *less* than 5, we connect $v$ with its 5 nearest nodes. If the number of nodes $u$ that meet this condition is *more* than 10, we connect $v$ with its 10 nearest nodes. Each edge $e_{uv}$ is weighted with $\omega_{uv} := D_{uv}$. Since intuitively we may expect that nearer nodes should have more influence in the construction of the histograms of unknown nodes, in the propagation functional we weight $\mathsf{S}_\varepsilon(\rho_v, \rho_u)$ with use $\exp(-\omega_{uv}/\sigma)$ or $1/\omega_{vu}$ suitably normalized.

Fig. 8: True barycenter (left), result of our algorithm (right)