[Reviews · NeurIPS 2019]

Reviewer 1



1. Overall, I like this work. I checked the proofs and believed that the proposed FW algorithm is solid. It would be better if authors can compare the proposed method with existing Wasserstein/Sinkhorn barycenter learning methods, e.g., [10, 12, 18, 48] and Ye, Jianbo, et al. "Fast discrete distribution clustering using Wasserstein barycenter with sparse support." IEEE Transactions on Signal Processing 65.9 (2017): 2317-2332. 2. In each iteration, the FW algorithm needs to solve a MINIMIZE problem in continuous or discrete domain and obtain x_{k+1} accordingly. Could authors provide more details about how to derive x_{k+1}? Is it based on finding a stationary point making the gradient of objective function zero? Or sampling some candidate points and finding that corresponding to the smallest objective value?

Reviewer 2



Overall, the paper is very clearly written with a unusually high level of mathematical sophistication and quality (e.g. compared to the usual NeurIPS submissions). I don't feel that the actual method is too novel, as Frank-Wolfe on measures have been considered previously. The strength and significance of the work is rather the particularly rigorous treatment, which puts such ideas on more firm theoretical grounds. **after rebuttal** My concerns mentioned in the "Improvements" sections were clearly addressed by the rebuttal, and I still vote for accepting the paper.

Reviewer 3



Pros: - This is a nice algorithm, as it doesn't require the support of the barycenter to be fixed beforehand and provides a simple way to control the number of support points. - The theorems proved might be useful in the other settings where optimal transport is used. The proofs in the appendix seem fine, although I should be honest that I have essentially just glanced over them. - The paper is well organized and is mostly easy to read. Cons: - What is the regularisation constant (epsilon) used for the experiments? It seems that the convergence essentially slows down exponentially as epsilon is decreased. - The rate of convergence seems to be independent of the number (m) of input distributions which you are averaging. Is this the case? Or probably it is hidden in the constants? - Besides the convergence analysis, it would be useful to compare the runtime of your algorithm with existing algorithms. Right now, nothing is said about the practical efficiency and how much it can be scaled. - What is the initialization strategy used for the barycenter in the experimental section? ---- UPDATE ----- Thanks for the rebuttal and doing the additional experiments. It addresses most of the concerns except for scalability, but I think that can hopefully be worked around in the future.

[Author Response · NeurIPS 2019]

We thank the reviewers for their comments. As requested by R1/R3 we first report an empirical comparison with
previous work. We then address reviewer's comments individually (due to space limits please zoom in the tiny figures).

| $\varepsilon$ | $B_\varepsilon$ | | Time (s) | | Tuning Time (s) | |
|---|---|---|---|---|---|---|
| | Ours | AM [12] | Ours | AM [12] | Ours | AM [12] |
| .01 | $.476 \pm \sigma$ | $.477 \pm \sigma$ | $13 \pm 1$ | $12 \pm 1$ | – | $482 \pm 20$ |
| .005 | $.434 \pm \sigma$ | $.436 \pm \sigma$ | $16 \pm 1$ | $15 \pm 1$ | – | $557 \pm 14$ |
| .001 | $.388 \pm \sigma$ | $.401 \pm \sigma$ | $20 \pm 2$ | $7 \pm 2$ | – | $242 \pm 8$ |

Figure 1

| m | n | |
|---|---|---|
| | 100 | 1000 |
| 10 | $29 \pm 4$ s | $33 \pm 6$ s |
| 50 | $8 \pm 1$ min | $9 \pm 1$ min |
| 100 | $15 \pm 1$ min | $24 \pm 2$ min |

Table 2: Time to reach relative improvement $10^{-4}$ of $B_\varepsilon$ ($\varepsilon = 0.01$).

Table 1: Comparison of our algorithm and [12] on Ellipses (same setting in the paper). $\sigma < 10^{-4}$.

**Comparison:** we focus on [12,18] since they consider entropic regularization similarly to our setting. Since no code
was available, experiments are based on our implementations of [12,18] (run on a Nvidia Tesla M40).
*Alternating Minimization (AM) [12]:* AM iteratively optimizes the support points and weights of the barycenter.
Table 1 reports the value of the objective functional $B_\varepsilon$ and the running times (run until relative improvement of
$B_\varepsilon$ was $< 10^{-3}$) on the "30 ellipses dataset" (see our paper) for $\varepsilon = 0.01, 0.005, 0.001$. We note that while a
single run of AM is slightly faster, it exhibits worse performance (in particular as $\varepsilon$ decreases). Moreover, dif-
ferently from our method, optimization in [12] requires tuning multiple parameters (e.g. step-size), leading to
significantly longer times. We run AM with a budget of 500 support points. Our method stops at $\sim 300$ support points.
*Decentralize Barycenters [18]:* similarly to our method, [18] can compute
barycenters of continuous measures (via sampling), *but* the barycenter's sup-
port points are *fixed a priori* and only the weights are computed. Moreover
since [18] minimizes a different objective functional (it does not consider
the *unbiased* formulation of the Sinkhorn divergence), here we focus on
a qualitative comparison. We compute the barycenter of 5 bidimensional
Gaussian measures $\mathcal{N}(m, \sigma^2 Id_2)$, with m randomly sampled in $[0,1]^2$ and
$\sigma^2$ in $[0,1]$. We set $\varepsilon = 0.01$ for both methods. For [18] we used Alg. 2
with complete agents' graph and a $50 \times 50$ support grid in $[0,1]^2$. Fig. 2
(left) shows how barycenters evolve over time. Our method appears to
better capture the properties of the target barycenter, converging faster
towards its solution. This is also reflected by the decreasing rate of the two
corresponding measure of convergence for the two methods Fig. 2 (Right).

Figure 2: Evolution of the barycenter of 5 Gaussians computed by [18] (Top row) and our algorithm (Bottom) over time. (Right column) distance to consensus (see [18]) and the $B_\varepsilon$ functional (with markers at 10, 30, 120 sec).

**R1 1.** We thank the reviewer for the additional reference, which we will add to the paper. **2.** There are several options
for the MINIMIZE routine: for $\mathcal{X}$ finite (e.g., images, such as our experiments on ellipses and k-means), we evaluate the
function at all points of $\mathcal{X}$ (in a vectorized way) and select the minimizer by a sorting algorithm. If $\mathcal{X}$ is a continuous
domain, we rely on first order methods (e.g. Gradient Descent) applied in parallel to multiple starting points. For
instance, in our experiments on Gaussians, we used the python `scipy.optimize` routine as a plugin optimizer.
**R2 1.** We thank R2 for the reference "Entropic regularization of continuous optimal transport problems". We note that
this work studies regularization with *negative entropy of the transport plan* $\pi$, whereas in our problem (1) the regularizer
is the *Kullback-Leibler* between $\pi$ and $\alpha \otimes \beta$. These two problems are not equivalent, e.g., if $\alpha$ or $\beta$ are not absolutely
continuous wrt Lebesgue. In our case, existence of solutions is a consequence of [28], which studies DAD problems in
*continuous settings*. Indeed, existence of maximizers for (2) is equivalent to existence of solutions to the optimality
equation (4), which is a special case of a DAD problem: thus, existence of maximizers for (2) follows from [28, Thm 1].
We cited [28] on line 67 and also discussed this issue in detail in Appendix B.2 (see Cor B.6). **2.** The interpretation
of (2) as primal and (1) as dual is indeed the way to derive strong duality, since in this interpretation the qualification
conditions hold (see e.g., the proof of Thm. 3.2 in [8]). However, problem (2) can also be seen as the dual of (1) when
the involved spaces are endowed with the weak topologies (this requires formulating the Fenchel-Rockafellar duality in
locally convex spaces [8, Thm A.1]); this follows the convention used in optimal transport literature. **3.** The work [28]
is in infinite dimensional setting (see also the convergence of the Sinkhorn-Knopp algorithm in Appendix B.3.)
**R3 1.** The exponential slow-down wrt $\varepsilon$ reflects recent findings on the Sinkhorn divergence, where similar scaling was
observed for, e.g., its sample complexity [21]. However, as also reported in Table 1 above, in practice the slow-down
does not seem too severe. In the paper we used $\varepsilon = 0.001$, which typically yields visually good results. **2.** According to
Thm 3 the convergence rate depends on the Lipschitz constant of the gradient of the objective function. Since $\nabla B_\varepsilon$ is a
weighted sum of $m$ mappings with same Lipschitz constant and the weights $\omega_i$ sum to 1, the Lipschitz constant of $\nabla B_\varepsilon$
does not depend on $m$. **3.** Scaling: we computed the barycenter of $m$ distributions with $n$ points each (obtained by
randomly displacing and sampling from the 2D distribution in Fig. 1). Table 2 shows the runtimes of our algorithm
as $m$ and $n$ vary. We observed that the main bottleneck of our method are the $m$ SINKHORNKNOPP computations
at each iteration (e.g. on average $\sim 94\%$ of the total time for $m = 100$ $n = 1000$). We plan to address this by $i$)
parallelizing with respect to the $m$ distributions and $ii$) adopting the very recent toolbox for Sinkhorn computation
`www.kernel-operations.io/geomloss/`, which can yield a $\sim 50$-$100\times$ speed-up to SINKHORNKNOPP. Such a
boost would allow us to consider larger scale settings. **4.** Initialization: a single Dirac Delta randomly sampled in $\mathcal{X}$.

[Meta-Review · NeurIPS 2019]

All reviewers agreed that this was a very good contribution and the rebuttal did a good job in addressing their concerns.